# EBM Life Cycle: MCMC Strategies for Synthesis, Defense, and Density Modeling

## Abstract

This work presents strategies to learn an Energy-Based Model (EBM) according to the desired length of its MCMC sampling trajectories. MCMC trajectories of different lengths correspond to models with different purposes. Our experiments cover three different trajectory magnitudes and learning outcomes: 1) shortrun sampling for image generation; 2) midrun sampling for classifier-agnostic adversarial defense; and 3) longrun sampling for principled modeling of image probability densities. To achieve these outcomes, we introduce three novel methods of MCMC initialization for negative samples used in Maximum Likelihood (ML) learning. With standard network architectures and an unaltered ML objective, our MCMC initialization methods alone enable significant performance gains across the three applications that we investigate. Our results include state-of-the-art FID scores for unnormalized image densities on the CIFAR-10 and ImageNet datasets; state-of-the-art adversarial defense on CIFAR-10 among purification methods and the first EBM defense on ImageNet; and scalable techniques for learning valid probability densities.

## 1 Introduction

Generative modeling of complex signals is one of the fundamental challenges of computational cognition. The importance of studying generative models is often justified by the Analysis-by-Synthesis hypothesis, which posits that human cognition is an interplay of bottom-up information from world states and top-down synthesis of imagined states. Powerful generative models could provide the foundation of the imaginative capabilities of future machine intelligence. One approach to generative modeling is to posit the existence of a density $q(x)$ of signals $x$ that generates data samples, and to learn $q(x)$ using a flexible model $p(x; \theta)$, where $\theta$ is a model parameter. This approach is taken by Energy-Based Models (EBMs) (Xie et al., 2016), normalizing flows (Kingma & Dhariwal, 2018), score-based models (Song & Ermon, 2019; 2020), and auto-regressive models (Oord et al., 2016), as well as by Variational Auto-encoders (VAEs) (Kingma & Welling, 2013) using a joint model $p(x, z; \theta)$. In this work we consider modeling $q(x)$ using an EBM density $p(x; \theta)$ for image signals $x$. See Appendix A for a brief review of Maximum Likelihood learning with an EBM.

The primary goal of most deep generative modeling is to generate realistic images. It is well-known that shortrun sampling with an EBM is an effective method for image generation, but synthesis results still lag behind GANs (Goodfellow et al., 2014), score models, and diffusion models (Ho et al., 2020). Other directions of EBM research include learning valid densities (Nijkamp et al., 2020), combining discriminative and generative learning via the Joint Energy Model (Grathwohl et al., 2020) and relatives, and using an EBM for defense against adversarial attacks (Hill et al., 2021). In this work, we focus on unconditional learning where EBMs are trained exclusively on unlabeled images. We explore the tasks of image synthesis, adversarial defense, and density modeling to examine a breadth of capabilities for the unconditional EBM. Image synthesis is a shared goal across all generative models, while defense and density modeling are tasks that are especially suitable for EBMs. See Appendices B and I for a thorough comparison of EBM and other generative models from the perspective of synthesis, defense, and density estimation.

Each task is naturally associated with a certain length of MCMC trajectory. Image synthesis is most effective with shortrun trajectories (about 20 to 200 steps) that can rapidly generate new images. Adversarial defense requires midrun trajectories (about 200 to 2000 steps) that can preserve the class

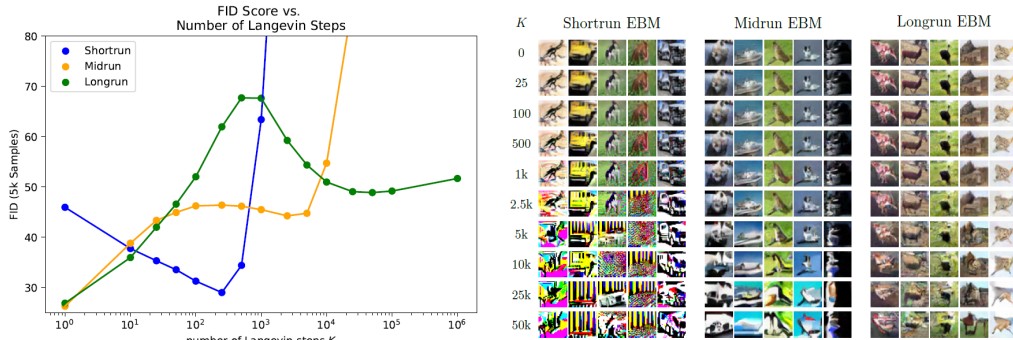

Figure 1: Illustration of the sampling trajectories that we study in this work. The shortrun samples are initialized from a generator that is trained in tandem with the EBM because the goal is self-contained synthesis. Midrun and longrun samples are initiated from a high-quality starting image obtained from a pre-trained SNGAN, and we study the ability of the EBMs to preserve the quality of the input image from defense and density estimation points of view. The plots show the FID score (Heusel et al., 2017) of 5,000 samples across Langevin steps. The shortrun samples improve on the generator initialization to achieve high-quality synthesis around 250 steps. The midrun samples achieve reasonably low FID in a critical range of about 2k steps where defense can be achieved. The longrun sample maintains reasonable synthesis across the entire trajectory, and much further. The shortrun and midrun samples eventually produce defective results outside of their tuned window of stability.

features while sampling removes adversarial signals. Density modeling requires longrun trajectories (50K steps or more) to ensure proper calibration of probability mass for the model steady-state. It is known that EBMs have a near universal tendency to learn a misaligned steady-state focusing on unrealistic images (Nijkamp et al., 2020), and there are very few existing solutions to correct this. We intuitively refer to the spectrum of trajectories as the life cycle of an MCMC sample, from youth through middle age to maturity. Young samples have the highest quality visual appearance, middle age samples are useful for securing classifiers, and mature samples represent grounded knowledge of the data density (or lack thereof in the widespread case of a misaligned steady-state). Figure 1 illustrates the sampling paths that we study in this work.

Ideally, a valid density estimator would also be a good synthesizer and a good defender. However, synthesis quality and density modeling are goals that tend to be at odds with each other. High-quality synthesis is easier to achieve using shortrun sampling with a defective density rather than long-run sampling with a stable density (Nijkamp et al., 2020). Unfortunately, models with high-quality shortrun synthesis lack the stability needed for defense. The EBM defense from Hill et al. (2021) advocates for the use of a valid density approximation to stabilize trajectories, but this requirement might be too strict given the difficulty of density estimation. Learning midrun sampling trajectories sufficient for defense is much more feasible for complex datasets such as ImageNet than full density estimation, even if longrun samples from the defensive model are not realistic. Our work is the first to explore this possibility. The overarching purpose of this paper is to discuss techniques for building sample paths from shortrun to longrun, with the hope that these techniques will eventually enable both high quality synthesis and defense to be accomplished with a model that is also a valid density. For now, the difficulty of valid density modeling leads us to restrict our focus to separate time scales of the sampling regime necessary for each task. Interestingly, EBM training naturally accommodates learning at different trajectory lengths.

Strategies for improving EBM learning beyond the standard framework (e.g. Xie et al. (2018); Du & Mordatch (2019); Nijkamp et al. (2020)) can broadly be divided into methods that focus on the initialization of MCMC samples (Xie et al., 2018; Gao et al., 2018; Nijkamp et al., 2019) and methods that focus on the ML learning objective (Yu et al., 2020b). Some works explore both (Gao et al., 2020a; Du et al., 2020). Our novel learning methods focus on MCMC initialization, and we retain the standard ML objective and use conventional network architectures. We introduce three new MCMC initialization strategies which are tailored to the three different trajectories lengths we explore. During training we exclusively use shortrun MCMC to ensure computational feasibility. Learning models with midrun and longrun trajectories is accomplished by simulating longer trajectories via well-

chosen initialization and optimizer annealing. Our initialization strategies are able to significantly improve the state-of-the-art across the tasks we investigate. We summarize our contributions below.

- We propose a hybridization of persistent (Tieleman, 2008) and cooperative (Xie et al., 2018) initialization to overcome limitations of each. The proposed method yields state-of-the-art FID scores for unconditional unnormalized image densities for the CIFAR-10 and ImageNet datasets. The ImagetNet results surpass GANs trained with similar resources. See Section 2.

- We show that persistent initialization with appropriately tuned rejuvenation from in-distribution states can be used to train EBMs with stable trajectories of several thousand MCMC steps. This allows us to extend the method of Hill et al. (2021) to obtain state-of-the-art purification-based defense for CIFAR-10 and to scale the EBM defense to ImageNet. See Section 3.

- We propose a method for principled density estimation that allows incorporation of a rejuvenation step for persistent states. Incorporating rejuvenation allows us to learn well-formed EBM densities at a greater scale than previously possible. See Section 4.

Our initialization methods will primarily build upon persistent (Tieleman, 2008; Du & Mordatch, 2019) and cooperative (Xie et al., 2018) initialization. All of our methods will use a generator network as the source of rejuvenation for persistent states. Our shortrun experiments will learn the generator in tandem with the EBM so the synthesis process is self-contained, while our midrun and longrun experiments will use pretrained generators since we will apply sampling paths from in-distribution initial images rather than synthesizing from scratch.

## 2 HYBRID PERSISTENT COOPERATIVE LEARNING FOR IMAGE SYNTHESIS

In this section, we focus on on the conventional task of learning an EBM for high quality synthesis with shortrun sampling. We restrict our attention to achieving high quality synthesis without use of pretrained models, in contrast with works such as Che et al. (2020); Alayrac et al. (2019). Our method involves hybridizing persistent (Tieleman, 2008) and cooperative (Xie et al., 2018) initialization. We first briefly cover the strengths and weaknesses of these methods, then discuss our hybridization, and finally present experimental results.

### 2.1 MOTIVATION: LIMITATIONS OF PERSISTENT AND COOPERATIVE INITIALIZATION

A common framework for learning synthesis with an EBM is to use persistent initialization for MCMC samples with a certain rate of rejuvenation (e.g. 5% chance) from a noise distribution (Du & Mordatch, 2019). This approach can cause instability because shortrun samples used to update the model include a mix of higher-energy burn-in samples and lower-energy realistic images, which can destabilize training by increasing the variance of the gradient of the negative samples in (4). On the other hand, removing rejuvenation can decrease the quality of the learned images because persistent images often become stuck in local modes and develop defects that linger for many updates. Persistent banks also scale poorly to large datasets such as ImageNet because the bank cannot efficiently represent the diversity of the data. Initialization from a cooperative generator network is an appealing alternative because it could enable efficient in-distribution generation of highly diverse appearances. Using a generator for rejuvenation allows samples to begin much closer to the correct energy spectrum, thereby avoid the instability of noise rejuvenation. However, we observe a major limitation of cooperative learning that affects the generator ability to produce diverse initial states. In particular, a generator trained with the cooperative learning objective has difficulty breaking the symmetry of its activations. This happens because shortrun EBM samples are unable to provide novel diversity if the generator initialization already lacks diversity. This is illustrated in Figure 5. Learning quickly becomes unstable without auxiliary techniques such as batch normalization to break generator symmetry. The lack of diversity of shortrun EBM samples limits the results of cooperative learning even when training succeeds.

The strengths and limitations of these methods are complementary. Persistent initialization can reliably provide a diverse set of initial images that represent prior samples of several different model snapshots. Even if a single EBM update is biased to a certain image defect (e.g. MCMC samples are blue-tinged or too bright), samples across previous iterations have the correct diversity on average. On the other hand, the generator can quickly propose in-distribution states for rejuvenation

instead of relying on out-of-distribution noise samples or aggressively augmented persistent samples. Furthermore, training the generator in tandem with the EBM provides an efficient way of generating new samples from scratch after training that match the samples used during training. In contrast, models trained with persistent initialization often cannot produce good samples after training without an in-distribution initialization (Nijkamp et al., 2020). Learning a generator and EBM jointly enables self-contained and reproducible FID implementation to evaluate the learned model.

## 2.2 HYBRID PERSISTENT COOPERATIVE INITIALIZATION

We visualize our proposed initialization technique in Figure 2. The method features two banks of persistent states. One bank is for persistent images and the other bank is for persistent latent vectors. There will always be a one-to-one pairing between persistent images and persistent latents, meaning that paired states will be drawn from the banks, return to the banks, and experience rejuvenation at the same time. All samples are rejuvenated from the current generator. Other than the pairing and generator rejuvenation source, sampling from banks and rejuvenation with a fixed probability is conducted in the same way as persistent initialization. This pairing is necessary for the cooperative learning loss used to train the generator. Given a batch of paired samples $\{Z_i\}_{i=1}^n$ and $\{X_{i,0}^-\}_{i=1}^n$ drawn from the latent and image bank respectively and a generator network $g(z; \phi)$, we learn $\phi$ using gradient descent on the reconstruction loss

$$\mathcal{L}(\phi) \propto \sum_{i=1}^n \|g(Z_i; \phi) - T(X_{i,0}^-; \theta)\|_2^2 \tag{1}$$

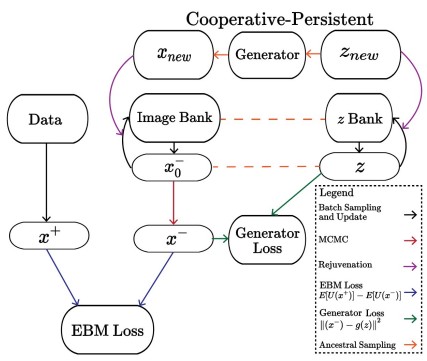

Cooperative-Persistent

Figure 2: Cooperative-persistent initialization uses paired latent and image states that a drawn from persistent banks to learn the EBM and generator.

where $T(X_{i,0}^-; \theta) = X_i^-$ represents a shortrun Langevin trajectory with the EBM $p(x; \theta)$. This loss can be derived in the ML framework as a way to teach the output of $g(z; \phi)$ to match the distribution of $p(x; \theta)$. A full implementation of cooperative learning requires an additional sampling process on $Z_i$, but we use a straight-through estimator (Bengio et al., 2013) and approximate latent sampling with the identity function as originally done in the official MATLAB implementation from Xie et al. (2018). We review further details of the cooperative learning loss in Appendix C.

Intuitively, the loss (1) encourages the output of the generator to match the outcome of the Langevin sampling process (5). Cooperative learning always uses $X_{i,0} = g(Z_i; \phi)$, which corresponds to rejuvenating with probability 1 in our method. This is the root of the limitation of cooperative learning. If $g(z; \phi)$ has little diversity, $T(g(z; \phi); \theta)$ will also have little diversity. This causes extreme oscillation in both the EBM and generator output as $p(x; \theta)$ attempts to cover the modes of $q(x)$ with shortrun samples from low diversity initialization. Despite oscillation in appearance across $\phi$, generator samples tend to remain nearly identical for any fixed $\phi$, thereby perpetuating the instability and preventing further learning. By drawing samples from the image bank, we are effectively choosing $X_{i,0} = T'(g(z; \phi'))$ where $\phi'$ represents a past generator parameter and $T'(x)$ represents the composition of Langevin sampling with past EBMs. This allows us to learn $\phi$ using appropriately diverse initializations spanning samples of several past models. We observe that this simple adjustment has a dramatic effect for improving EBM learning with generator initialization.

## 2.3 EXPERIMENTS: IMAGE SYNTHESIS WITH SHORTRUN MCMC

We present the results of our new learning process applied to the CIFAR-10, Celeb-A, and ImageNet benchmark datasets, using the image sizes $32 \times 32$, $64 \times 64$, and $128 \times 128$ respectively. A sketch of the hybrid learning algorithm is given in Appendix D. Besides standard deep learning techniques for stable optimization, our framework is fully described by the ML objective and MCMC initialization

Table 1: Comparison of FID scores among representative generative models. (*=EBM, †=conditional)

**CIFAR-10** $32 \times 32$

| Model | FID |
|---|---|
| **Ours*** | **22.1** |
| Du et al. (2020)* | 25.1 |
| Grathwohl et al. (2021)* | 27.5 |
| Xie et al. (2018)* | 33.6 |
| Gao et al. (2020a)* | 37.3 |
| Grathwohl et al. (2020)* | 38.4 |
| Du & Mordatch (2019)* | 40.6 |
| Yu et al. (2020a)*† | 30.9 |
| Du & Mordatch (2019)*† | 37.9 |
| Ho et al. (2020) | **3.2** |
| Song & Ermon (2020) | 10.9 |
| Brock et al. (2019) | 14.7 |
| Miyato et al. (2018) | 21.7 |

**Celeb-A** $64 \times 64$

| Model | FID |
|---|---|
| **Ours*** | 16.3 |
| Miyato et al. (2018) | **5.7** |
| Song & Ermon (2020) | 10.2 |
| Han et al. (2019)* | 31.9 |

**ImageNet** $128 \times 128$

| Model | FID |
|---|---|
| **Ours*** | **38.9** |
| Chen et al. (2019) | 43.9 |
| Lee et al. (2021) | 58.9 |
| Miyato et al. (2018) | 65.7 |
| Miyato et al. (2018)† | 27.6 |
| Du & Mordatch (2019)*† | 43.7 |

above. We use the SNGAN (Miyato et al., 2018) architectures for all models, where EBMs use the SNGAN discriminator with no normalization. The generator has batch normalization for the CIFAR-10 and Celeb-A experiments only. Table 1 displays the FID scores achieved by our model in comparison with prior methods.

When calculating FID scores, samples are first initialized from the generator then updated with the EBM using a number of Langevin updates tuned to provide optimal synthesis quality. As mentioned before, an appealing aspect of EBM learning with generator initialization is the ease of generating new images from scratch, in contrast with persistent initialization. We use 50K samples with the official FID code from Heusel et al. (2017) to calculate all FID scores. In particular, our scores and framework are consistent with the GAN replication library from Lee & Town (2020). We publicly release the checkpoints, learning code, and FID code for each model. The checkpoints are representative of what is achievable within an ordinary run of the code provided. Our CIFAR-10 results show a significant improvement over prior EBM synthesis and contribute to closing the gap between EBMs and other generative models. Surprisingly, our ImageNet results surpass the results of GANs such as SNGAN (Miyato et al., 2018) and SSGAN (Chen et al., 2019) on unconditional synthesis using a similar magnitude of computational resources.

# 3 MIDRUN SAMPLERS FOR ADVERSARIAL DEFENSE

This section presents a method for learning EBMs that are capable of preserving the appearance of an in-distribution initial state across several thousand MCMC steps. Such models are useful for the purpose of adversarial defense. Our defense framework is based on the approach in Hill et al. (2021), which uses an EBM to defend an independent naturally trained classifier. Appendix E briefly reviews the EBM defense and compares this approach with other defense methods. We then present our proposed method for learning a defensive EBM (3), which is based on persistent initialization using a fixed pretrained generator as a source of rejuvenation. Finally, we apply our defense to achieve state-of-the-art performance for purification-based defense on CIFAR-10 and ImageNet.

One limitation of prior EBM defense is the reliance on persistent initialization with no rejuvenation to learn the defensive model. This is done to ensure that defensive sampling trajectories remain stable for arbitrary numbers of steps. However, removing rejuvenation from persistent learning has drawbacks discussed in Section 2.1. In particular, it becomes very difficult to learn meaningful EBMs for large and complex datasets such as ImageNet in this framework because the persistent bank cannot represent the diversity of the dataset and the quality of persistent images without rejuvenation quickly degrades. Methods for efficient learning of defensive EBMs at a greater scale are needed to extend the EBM defense to more realistic situations.

We propose to overcome this obstacle by building on the observation that fully stable sampling paths are not required for successful defense. While the defense from Hill et al. (2021) uses EBMs with stable sampling for 100K steps or more, the defense results require less than 2000 steps. A natural question is whether it is possible to learn stable MCMC trajectories for only a predefined midrun range to achieve the defensive benefits without fully stabilizing samples over longrun trajectories. Defining a learning procedure to obtain such models is the goal of this section. Efficiently learning EBMs with stable midrun trajectories allows us to scale up the EBM defense to significantly more challenging domains.

### 3.1 In-Distribution Rejuvenation for Learning a Defensive EBM

The initialization method for our midrun sampler is similar to standard persistent initialization with the adjustment that persistent states are rejuvenated from a frozen pretrained generator rather than noise. We use a generator in our experiments so that the EBM and generator could be used after training to sample from scratch. This choice is not necessary and rejuvenation from data samples, or another efficient in-distribution initialization, is also effective for learning defensive EBMs. We note that MCMC initialization from a trained generator or from data samples is not explored in recent work because the goal of most current EBM learning is image synthesis, which becomes trivial if EBM trajectories are always initialized from high-quality samples. From the perspective of synthesis our learning process is nearly invisible but from the perspective of defense its utility becomes concrete.

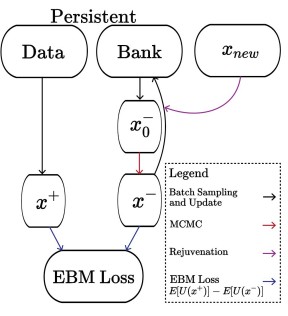

Figure 3: Persistent initialization. Positive samples are from data samples and negative samples are MCMC samples initialized from a batch from an image bank. Some states are randomly rejuvenated.

Learning an EBM for defense involves tuning the length of the sampling trajectory via the number of shortrun training steps and the rejuvenation rate, and tuning the annealing schedule of the EBM optimizer. Given a desired number of MCMC steps $K_{\text{def}}$ for a defensive update and a shortrun trajectory $K \approx 100$, we simply set the rejuvenation rate to $p_{\text{rejuv}} = K/K_{\text{def}}$ to ensure that on average samples will travel $K_{\text{def}}$ steps before rejuvenation. While in practice we use $K_{\text{def}} = 2000$ and $p_{\text{rejuv}} = 0.05$, we have found that this method can yield stable paths for at least $K_{\text{def}} = 50K$ MCMC steps when $p_{\text{rejuv}}$ is low.

Initialization alone is insufficient to stabilize MCMC pathways when model weights are changing quickly. Using a low learning rate late in training is a key aspect of stabilizing sampling paths (Nijkamp et al., 2020; Hill et al., 2021). Intuitively, if the EBM optimizer has a sufficiently low learning rate then MCMC trajectories in the persistent image bank can function as approximate trajectories from the current model, since weights change very little as the persistent states are updated. By annealing in tandem with our initialization, we are effectively using midrun trajectories of length $K_{\text{def}}$ initialized from the generator to update the EBM while we are actually using shortrun trajectories of length $K$ from the persistent bank. Annealing is a crucial component for stabilizing both midrun and longrun trajectories. Without annealing, sampling paths are not able to maintain realism for large $K_{\text{def}}$.

### 3.2 Experiments: Defending Natural Classifiers with an EBM

We train our EBMs using the persistent initialization described above in tandem with a pretrained generator on both CIFAR-10 and ImageNet. We use the same SNGAN models as before for our CIFAR-10 experiments, with the exception that the generator is pretrained instead of learned. For our ImageNet experiments, we use the BigGAN (Brock et al., 2019) discriminator architecture for our EBM modified for input size $224 \times 224$ and a pretrained BigGAN Generator. Our naturally trained classifier $f(x)$ is a pretrained WideResNet 28-10 (Zagoruyko & Komodakis, 2016) for CIFAR-10 and a pretrained EfficientNetB-7 architecture (Tan & Le, 2020) for ImageNet.

We evaluate our models using the attack gradient (8) and Algorithm 2. For CIFAR-10 we use $K_{\text{def}} = 500$ Langevin steps with $l_\infty$ adversarial parameters $\varepsilon = \frac{8}{255}$ and $\alpha = \frac{2}{255}$, where $\varepsilon$ is the size of the $l_\infty$ ball and $\alpha$ is the gradient step size. For ImageNet we used $K_{\text{def}} = 200$ Langevin steps for defense with $l_\infty$ adversarial parameters $\varepsilon = \frac{2}{255}$ and $\alpha = \frac{1}{255}$. We attack ImageNet for 50 attacks

Table 2: Defense vs. whitebox attacks with $l_\infty$ perturbation $\varepsilon = 8/255$ for CIFAR-10.

| Defense | $f(x)$ Train Ims. | $T(x)$ Method | Attack | Nat. | Adv. |
|---|---|---|---|---|---|
| **Ours** | Natural | Langevin | BPDA+EOT | 0.866 | **0.676** |
| (Hill et al., 2021) | Natural | Langevin | BPDA+EOT | 0.8412 | 0.5490 |
| (Song et al., 2018) | Natural | Gibbs Update | BPDA | 0.95 | 0.09 |
| (Srinivasan et al., 2019) | Natural | Langevin | PGD | – | 0.0048 |
| (Yang et al., 2019) | Transformed | Mask + Recon. | BPDA+EOT | 0.94 | 0.15 |
| (Carmon et al., 2019) | Adversarial | – | PGD | 0.897 | 0.625 |
| (Zhang et al., 2019) | Adversarial | – | PGD | 0.849 | 0.5643 |
| (Shafahi et al., 2019) | Adversarial | – | PGD | 0.859 | 0.4633 |
| (Madry et al., 2018) | Adversarial | – | PGD | 0.873 | 0.458 |

steps across 10K val samples. For CIFAR-10 we perform the same number of attacks across 4K validation samples.

Table 3: Defense vs. $l_\infty$ whitebox attacks for ImageNet.

| Defense | $f(x)$ Train Ims. | $\varepsilon$ | Nat. | Adv. |
|---|---|---|---|---|
| **Ours** | Natural | $\frac{2}{255}$ | 0.684 | 0.418 |
| (Wong et al., 2020) | Adversarial | $\frac{2}{255}$ | 0.609 | 0.4339 |
| (Shafahi et al., 2019) | Adversarial | $\frac{2}{255}$ | 0.644 | 0.4339 |
| (Qin et al., 2019) | Adversarial | $\frac{4}{255}$ | 0.822 | 0.427 |
| (Xie et al., 2021) | Adversarial | $\frac{4}{255}$ | 0.822 | **0.586** |

On CIFAR-10, we surpass the robustness of the existing EBM defense using a much more reliable learning framework. The importance of midrun learning is clearly demonstrated by our successful application of EBM defense to ImageNet at the resolution $224 \times 224$. The robustness of a naturally trained classifier secured by our EBM is comparable with adversarial training. While the ImageNet results for EBM defense are not yet on par with state-of-the-art adversarially trained models, our experiments are an important proof of concept that the method can be scaled. See Appendix F for diagnostics and further discussion.

## 4 LONGRUN SAMPLING FOR DENSITY ESTIMATION

Our final objective is to introduce scalable tools for learning a valid image density with an EBM. Nijkamp et al. (2020) revealed that, in the absence of careful implementation, EBM learning always results in an unexpected outcome where steady-state samples from the learned density $p(x; \theta)$ have an oversaturated and unrealistic appearance that differs drastically from shortrun samples used during learning. This outcome affects all EBMs that are not specifically trained to overcome this defect, as well as related models such as Score Matching (Song & Ermon, 2019; 2020) as shown in Figure 8. To our knowledge, across different generative models, the only way to learn a well-formed potential energy surface approximating a complex density $q(x)$ that is compatible with efficient MCMC sampling is using an EBM.

Despite the theoretical formulation of the EBM as a potential surface, the problem of principled density estimation has received relatively little attention. Successful image synthesis is sometimes used as misleading evidence of successful density estimation, but we emphasize these outcomes not equivalent. In this section, we address the lack scalable methods to learn valid densities of complex signals by introducing an MCMC initialization can incorporate rejuvenation while still simulating extremely long trajectories. Learning a valid density is a fundamental computational problem that is important in its own right, and we further hope that our learning method leads to EBM clustering techniques based on the potential energy basins of a well-formed density (Nijkamp et al., 2020).

### 4.1 INCORPORATING REJUVENATION IN DENSITY ESTIMATION

Prior work has suggested that persistent learning is the most effective method for learning a valid EBM density. Furthermore, works that learn a valid density have avoided rejuvenation because

the incorporation of newly rejuvenated samples into the persistent bank ensures that EBM updates will always include samples that are not at the steady-state. However, persistent learning without rejuvenation has shortcomings mentioned in Section 3. We present hypothesized conditions for learning a valid density that motivate the design of our MCMC initialization:

- After a certain point in training, all samples used to update the EBM must be approximate steady-state samples of the current model $p(x; \theta)$.

- Persistent samples that are newly rejuvenated (up to about 50K Langevin steps since rejuvenation, and possibly many more) cannot be approximate steady-state samples for any known rejuvenation sources, including data, generators, and noise.

- Persistent samples that have undergone sufficiently many *lifetime* Langevin updates for a model whose weights are changing very slowly can be approximate steady-state samples.

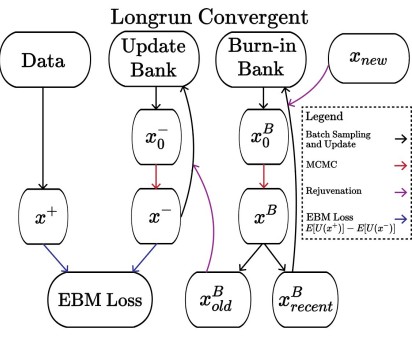

Figure 4: Visualization of our longrun initializatin procedure. Newly rejuvenated samples must remain the the burnin bank until they have approach the model steady-state, at which point they move to the update bank to be used for model gradients.

Both the second and third point are corroborated by prior work (Nijkamp et al., 2020; Hill et al., 2021) as well as our own observations. The third point means that persistent states updated with shortrun Langevin can eventually act as longrun Langevin samples if the optimizer learning rate is small, because the EBM samples in previous timesteps are essentially samples from the current EBM.

Learning a valid density that includes rejuvenation while satisfying the conditions above requires separating the newly rejuvenated samples from samples that are used to update the EBM. This leads us to introduce two persistent image banks: one for newly rejuvenated samples, and one for samples that will be used to update the EBM. Samples in the newly rejuvenated bank that have been updated sufficiently many times will eventually replace samples from the bank used to update the EBM, at which point newly rejuvenated states will be added to the burn-in bank. Figure 4 shows a visualization of the MCMC initialization method. As in Section 3, we will use a pre-trained generator to efficiently obtain high-quality rejuvenated samples so that the generation process is fully synthetic, although data samples could be used as well. Our goal is to preserve the sample quality for an arbitrary number of MCMC steps. We note that this is a sufficient but not necessary condition for learning a valid EBM density. Nonetheless, stable sampling is an important step towards rigorous probabilistic EBMs.

Even with our improved initialization, we find that extremely longrun trajectories of 1 million or more MCMC steps still tend to oversaturate, although to a much lesser degree. To further stabilize the appearance of extremely longrun Langevin samples, we include prior energy terms in the model. Our longrun EBMs have the form

Table 4: FID for 5K samples after 100K Langevin and 1M Langevin steps. FID remains stable over long trajectories.

| Data | 100K | 1M |
|---|---|---|
| CIFAR-10 | 49.2 | 51.7 |
| Celeb-A | 37.4 | 45.9 |
| ImageNet | 82.3 | 77.8 |

$$U_{\theta_0, \sigma}(x; \theta) = U'(x; \theta) + U'(x; \theta_0) + \frac{1}{2\sigma^2} \|x\|_2^2 \qquad (2)$$

where $U'(x; \theta)$ is the model whose weights are updated, $U_0(x; \theta_0)$ is a prior EBM with fixed weights $\theta_0$ and $\sigma$ is a parameter controlling the strength of a Gaussian prior. We used a prior EBM in a shortrun manner. The role of the prior EBM is to provide some stability but also to provide a tendency to oversaturate at longer trajectories so that the current EBM learns to correct oversaturation. The Gaussian prior is meant to discourage unbounded activations outside of the image hypercube. Further discussion is in Appendix H. We find that including both of these terms significantly improves the ability to learn quality synthesis over long trajectories.

## 4.2 EXPERIMENTS: EBM DENSITY ESTIMATION

In these experiments, we learn EBM models that have stable MCMC trajectories for an arbitrary number of steps. We apply the longrun learning method in Algorithm 4 to CIFAR-10, Celeb-A, and

ImageNet at the resolutions $32 \times 32$, $64 \times 64$, and $64 \times 64$ respectively. Pre-trained SNGANs are used to rejuvenate images for CIFAR-10 and Celeb-A, while a resized images from a pre-trained BigGAN are used to rejuvenate ImageNet samples. We require that samples have been updated for 50K to 75K Langevin steps since the last rejuvenation before states are moved from the burn-in bank to the update bank. Our EBMs all use the discriminator architecture of the SNGAN model without spectral normalization. We use pretrained midrun samples for our prior EBM terms in the learned energy (2). See Appendix K for an algorithm sketch. Figure 1 shows the evolution of FID score over Langevin steps starting from the generator network for our CIFAR-10 model. Table 4 shows the FID scores of 5K samples for each dataset. Visualizations of longrun samples with 100K steps and extremely longrun samples using 1 million Langevin steps are shown in Appendix M. See Appendix J for a discussion of our evaluation in comparison to log likelihood.

## 5 RELATED WORK

**Energy-Based Models.** Early forms of EBMs include the exponential family distribution, the FRAME model (Zhu et al., 1998) and Restricted Boltzmann Machines (Hinton, 2002). Recent work has introduced the EBM with a ConvNet potential (Xie et al., 2016; Du & Mordatch, 2019). This dramatically increased the learning capacity of the model which led to many follow-up works on image synthesis (Gao et al., 2018; Lee et al., 2018; Nijkamp et al., 2019), adversarial robustness (Hill et al., 2021), and joint learning of discriminative and generative models (Grathwohl et al., 2020). Several works investigate training and EBM in tandem with an auxilary model. Kim & Bengio (2016) train an EBM and generator and tandem without MCMC by using samples from the generator as direct approximations of the EBM density and training the generator using a variational objective. A similar approach is explored by Grathwohl et al. (2021). Cooperative learning (Xie et al., 2018) trains the EBM and generator by using the generator to initialize samples needed to train the EBM and uses reconstruction loss between generator and EBM samples to learn the generator. Gao et al. (2020a) learn an EBM using Noise Contrastive Estimation with an auxiliary flow model. Xiao et al. (2021) use a pretrained VAE to facilitate EBM learning. Our work builds on cooperative learning by identifying and resolving symmetry breaking problems in early training, leading to state-of-the-art EBM synthesis for unconditional ImageNet. Despite the formulation of the EBM as an unnormalized density, it has been shown that most EBMs have strong misaligned steady-state distributions (Nijkamp et al., 2020). Our work introduces new methods to learn a model with correct steady-state alignment.

**Adversarial Robustness.** Adversarial Training (AT) (Madry et al., 2018), which trains a classifier using PGD-generated adversaries, is the most popular and studied adversarial defense. Many variations and improvements have been introduced, including optimizing the training loop by recycling gradients of past adversaries (Shafahi et al., 2019), combining single step FGSM with random initialization to achieve similar robustness Wong et al. (2020), learning with auxiliary unlabeled data Carmon et al. (2019), local linearization (Qin et al., 2019), and the use of smooth activation functions Xie et al. (2021). An alternative approach to adversarial training involves the use of preprocessing transformations. Randomized smoothing (Cohen et al., 2019) and related methods (Salman et al., 2020) add noise to the input signal to remove adversarial signals. Many other preprocessing defenses have been proposed (Guo et al., 2018; Song et al., 2018; Yang et al., 2019), but nearly all of these methods can be broken by adaptive attacks that are aware of the preprocessing method (Athalye et al., 2018). A notable exception is the EBM defense (Hill et al., 2021), which uses midrun MCMC trajectories to purify images. We ease the restriction of learning EBMs with stability for arbitrary MCMC runs in the EBM defense by introducing a midrun sampler that enables faster learning of defensive EBMs and allows the EBM defense to scale to more complex datasets.

## 6 CONCLUSION AND FUTURE WORK

We have described three unique MCMC initializations for EBM using different sampling trajectories: shortrun for synthesis, midrun for defense, and long-run for density estimation. Furthermore, we have elaborated on different MCMC initialization strategies used to stabilize these models for different sampling lengths. We have demonstrated the flexibility of these mechanisms by using similar architectures, data, and training platforms to create different EBMs for different applications. We hope that future research incorporates these new training initialization schemes to improve their generative models for a wide variety of tasks.

## 7 REPRODUCIBILITY

We have made our results reproducible by releasing the anonymized github repository `https://anonymous.4open.science/r/lifecycle-3346/`. All hyperparameters can be found in the configuration files, all checkpoints for the various EBMs are also under the release section and there is a README.MD describing how to use this repository to reproduce the results described herein. We will release a non-anonymous repository at a later date. All our results were trained on the standard CIFAR-10, ImageNet, and Celeb-A datasets which are publicly available. Our computational resources were primarily 5 TPUv2-8 machines and 5 TPUv3-8 machines made available by the Tensorflow Research Cloud (TRC) program sponsored by Google. These resources are widely available to researchers who apply to the program from accredited institutions. We are very grateful to Google for granting us access to the resources that made this work possible.

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

## A    Review of EBM Learning and MCMC Initialization

We briefly review the main components of EBM learning following the standard method derived from works such as Hinton (2002); Zhu et al. (1998); Xie et al. (2016). Throughout our experiments, we will use this framework and focus on exploring different possibilities for MCMC initialization. A deep EBM has the form

$$p(x; \theta) = \frac{1}{Z(\theta)} \exp\{-U(x; \theta)\} \tag{3}$$

where $U(x; \theta)$ is a ConvNet with weights $\theta$ and $Z(\theta)$ is the intractable normalizing constant. Maximum Likelihood (ML) learning uses the objective $\arg\min_\theta D_{KL}(q(x)||p(x; \theta))$, which can be minimized using the stochastic gradient

$$\nabla \mathcal{L}(\theta) \approx \frac{1}{n} \sum_{i=1}^{n} \nabla_\theta U(X_i^+; \theta) - \frac{1}{n} \sum_{i=1}^{n} \nabla_\theta U(X_i^-; \theta) \tag{4}$$

where the positive samples $\{X_i^+\}_{i=1}^n$ are a set of data samples and the negative samples $\{X_i^-\}_{i=1}^n$ are samples from the current model $p(x; \theta)$. To obtain the negative samples for a deep EBM, it is common to use MCMC sampling with the $K$ steps of Langevin equation

$$X^{(k+1)} = X^{(k)} - \frac{\eta^2}{2} \nabla_{X^{(k)}} U(X^{(k)}; \theta) + \eta Z_k, \tag{5}$$

where $\eta$ is the step size and $Z_k \sim N(0, I)$. The Langevin trajectories are initialized from a set of states $\{X_{i,0}^-\}_{i=1}^n$ obtained from a certain initialization strategy. The method of MCMC initialization determines many of the properties of the learned model.

There are a number of different methods of MCMC initialization for EBM learning. Early Restricted Boltzmann Machines use data samples as the initial states in every iteration (Hinton, 2002). Persistent initialization (Zhu et al., 1998; Tieleman, 2008) maintains a bank of MCMC samples from previous model updates as the source of MCMC states in the current iteration. Persistent states can be rejuvenated with a certain probability instead of returning to the image bank. Potential sources for rejuvenation include noise samples (Du & Mordatch, 2019), data augmentation applied to persistent states (Du et al., 2020), data samples, or samples from a generator network. Cooperative learning initializes MCMC states from a generator learned in tandem with an EBM (Xie et al., 2018). The Multi-grid model (Gao et al., 2018) initializes samples from lower resolution images. Noise initialization is explored by Nijkamp et al. (2019). The VAEBM (Xiao et al., 2021) initializes samples from a pretrained VAE generator. EBMs can learn stable shortrun synthesis from virtually any initialization distribution as long as the initialization distribution remains fairly consistent across training updates. EBM sampling can produce realistic images after training when the testing initialization closely matches the training initialization, while synthesis can fail for test initializations unseen during training. (Nijkamp et al., 2020).

## B    Comparison of EBM and Other Generative Models

In this section, we briefly compare the EBM with other kinds of generative models. Besides the probabilistic model families mentioned in the introduction, we consider two other popular approaches. One approach is to design an objective that promotes realistic synthesis by producing fake samples and distinguishing these from real samples, as done by the Generative Adversarial Network (GAN) model (Goodfellow et al., 2014) and its many variations. Another approach comes from recent diffusion models that define a sequence of intermediate distributions to transport a signal between a reference density $p_0(x)$ and $q(x)$ (Ho et al., 2020; Song et al., 2020).

Among probabilistic models, the synthesis quality is typically the highest for score matching models that only try to learn distribution gradients and typically lower for models like normalizing flows and auto-regressive objectives that model the full normalized density. EBM synthesis usually lies somewhere in between. Score-matching does not require MCMC sampling during learning and as a consequence learning tends to be more straightforward than learning for EBMs. Yet we believe that MCMC sampling is not yet fully optimized for EBM learning and that EBMs can match or surpass score-matching models. Our shortrun initialization method is an effort in this direction.

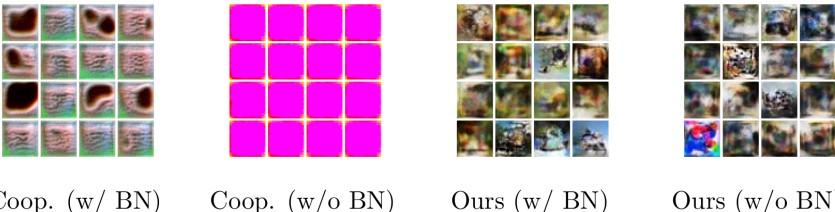

| Coop. (w/ BN) | Coop. (w/o BN) | Ours (w/ BN) | Ours (w/o BN) |

Figure 5: Comparison of cooperative learning (Xie et al., 2018) and our hybrid cooperative-persistent learning using appearance of shortrun samples after 500 updates of the EBM using a generator with and without batch norm. The cooperative models have difficulty achieving diversity in the shortrun samples because the EBM is unable to significantly change the appearance of initial generator images. In fact, we find cooperative learning is not possible at all without generator batch normalizatin. Our hybrid initialization increases the diversity of shortrun samples by including persistent samples from previous EBM updates. This accelerates learning and dramatically improves synthesis results.

GANs and diffusion models differ fundamentally from probabilistic models of the form $p(x; \theta)$ in the sense that the former are explicitly formulated from the perspective of synthesis while the latter are formulated for density modeling and synthesis is a byproduct. In the literature, the former types of models are sometimes referred to as *generative* models (in a more specific sense of the term than typical) while the latter are called *descriptive* models (Guo et al., 2003; Zhu, 2003). Generative models map one set to another (e.g. a generator mapping a latent vector to an image), while descriptive models define a concept in terms of the set of states that have a certain descriptive property (e.g. steady-state samples of an EBM). A descriptive model can be used for generative purposes (e.g. Langevin sampling with an EBM) but this is not the direct goal of descriptive learning.

From this perspective, it is perhaps unsurprising that generative methods such as GANs and diffusion models surpass EBMs in terms of synthesis, since that the model formulations are geared directly towards generation. Nonetheless, we believe that EBMs have unique properties that distinguish them from generative models and other descriptive models. In particular, EBM learning offers a control of the MCMC sampling paths from a learned model that $p(x; \theta)$ is, to our knowledge, is unobtainable under other current frameworks. Unlike GAN models which map a trivial distribution to a complex signal distribution without actually describing the signal distribution, EBMs directly model the complex signal distribution via a potential energy surface. Normalizing flow models are too large for efficient MCMC sampling. VAE models cannot be efficiently marginalized. Score-based models exhibit the same oversaturation behavior as miscalibrated EBMs when sampling at a constant noise level for many steps (see Figure 8), and one cannot adjust MCMC stability during learning as we do for EBMs, because sampling is not used when training a score model. Diffusion-based methods do not learn an approximation $p(x; \theta)$, but rather can be formulated as learning a conditional probability $p(x|x'; \theta)$ that describes a state $x$ given a state $x'$ a rung above or below $x$ in the ladder of distributions between $p_0(x)$ and $q(x)$ (Gao et al., 2020b). Auto-regressive models depend on a certain sequence of components for generation, and are not compatible with joint updates of all pixels. On the other hand, MCMC sampling with an EBM is a natural and relatively efficient process. Uniquely among all methods, the desired stability and length of MCMC sampling trajectories for an EBM can easily be controlled by adjusting the sampling phase of Maximum Likelihood (ML) learning. This can yield model capabilities not obtainable by other kinds of unsupervised models.

## C  COOPERATIVE LEARNING FOR GENERATOR NETWORK

In this section we briefly review the Maximum Likelihood framework for cooperatively learning the generator in tandem with the EBM (Xie et al., 2018). The generator model assumes a joint distribution $(X, Z)$ of images $X$ and latent signals $Z$ given by

$$Z \sim \mathrm{N}(0, I) \quad \text{and} \quad X|Z \sim \mathrm{N}(g(Z; \phi), \tau^2 I)$$

for a generator network $g$ with weights $\phi$ and a Gaussian parameter $\tau$ controls the spread of $X$ around $g(Z; \phi)$. Given i.i.d. pairs $\{X_i, Z_i\}_{i=1}^n$, the observations have a joint negative log probability

$$-\log p(\{X_i, Z_i\}_{i=1}^n; \phi) = \sum_{i=1}^n \left[ \|X_i - g(Z_i; \phi)\|_2^2 / (2\tau^2) + \frac{1}{2}\|Z_i\|_2^2 \right] + C.$$

In practice, the $X_i$ are shortrun MCMC samples from an EBM and the $Z_i$ are unobserved. The full formulation of cooperative learning involves latent MCMC using the conditional distribution $p(Z|X_i; \phi)$ to infer each $Z_i$. In practice, since the $X_i$ are shortrun samples that are paired with $Z_i$ through the process of rejuvenation, one can roughly approximate the latent MCMC process with an identity function and obtain a gradient (1). We find that this works as well or better than including a latent sampling step on $Z$, and it is consistent with the original implementation from Xie et al. (2018).

Figure 5 illustrates the importance of including a persistent image bank in the cooperative learning framework. Shortrun samples from the generator in standard cooperative learning have low diversity and it is difficulty for the generator to break the symmetry of its activations early in training. This can lead to catastrophic instability without auxiliary techniques like batch norm to help break the symmetry. Incorporating persistent states from prior EBM updates when updating the generator helps the shortrun distribution achieve a health diversity that enables stable and effective learning.

## D    COOPERATIVE-PERSISTENT ALGORITHM FOR SHORTRUN LEARNING

---
**Algorithm 1** Cooperative-Persistent Hybrid Learning
---
**Require:** Natural images $\{x_m^+\}_{m=1}^M$, EBM $U(x; \theta)$, generator $g(z; \phi)$ Langevin noise $\eta$, number of shortrun steps $K$, EBM optimizer $h_U$, generator optimizer $h_g$, initial weights $\theta_0$ and $\phi_0$, rejuvenation probability $p$, max update rounds $w$, number of training iterations $T$.
**Ensure:** Learned weights $\theta_T$ and $\phi_T$.
 Initialize bank of random latent states $\{Z_i\}_{i=1}^N$.
 Initialize image bank $\{X_i^-\}_{i=1}^N$ from generator using $X_i^- = g(Z_i; \phi_0)$
 **for** $1 \le t \le T$ **do**
  Select batch $\{\tilde{X}_b^+\}_{b=1}^B$ from data samples $\{x_m^+\}_{m=1}^M$.
  Get paired batches $\{\tilde{Z}_b\}_{b=1}^B$ and $\{\tilde{X}_{b,0}^-\}_{b=1}^B$ from $\{Z_i\}_{i=1}^N$ and $\{X_i^-\}_{i=1}^N$.
  Update $\{\tilde{X}_{b,0}^-\}_{b=1}^B$ with $K$ Langevin steps (5) to obtain negative samples $\{\tilde{X}_b^-\}_{b=1}^B$.
  Get learning gradient $\Delta_U^{(t)}$ using (4) with samples $\{\tilde{X}_b^+\}_{b=1}^B$ and $\{\tilde{X}_b^-\}_{b=1}^B$.
  Update $\theta_t$ using gradient $\Delta_U^{(t)}$ and optimizer $h_U$.
  Get learning gradient $\Delta_g^{(t)}$ using (1) with samples $\{\tilde{Z}_b\}_{b=1}^B$ and $\{\tilde{X}_b^-\}_{b=1}^B$.
  Update $\phi_t$ using gradient $\Delta_g^{(t)}$ and optimizer $h_g$.
  Rejuvenate each $\tilde{Z}_b$ from latent distribution with probability $p$.
  Also rejuvenate states $\tilde{Z}_b$ for which $\tilde{X}_b$ has been updated more than $w$ times.
  If $\tilde{Z}_b$ is rejuvenated, rejuvenate $\tilde{X}_b = g(\tilde{Z}_b; \phi_t)$.
  Return $\{\tilde{Z}_b\}_{b=1}^B$ to $\{Z_i\}_{i=1}^N$ and $\{\tilde{X}_b^-\}_{b=1}^B$ to $\{X_i^-\}_{i=1}^N$ by overwriting previous states.
 **end for**
---

## E    BACKGROUND ON EBM DEFENSE

The most popular method for adversarial defense is adversarial training (AT) (Madry et al., 2018; Wong et al., 2020; Shafahi et al., 2019) which aims to train a classifier to correctly predict adversarial samples within a small ball around a natural input. Another popular defense method is randomized smoothing (Cohen et al., 2019), which adds Gaussian noise to images to remove adversarial signals before classification. Both of these approaches modify classifier training. Although many methods have been proposed to secure a naturally trained classifier, most have been broken by stronger attacks. A recent method that has been shown to secure a natural classifier is Langevin sampling with an EBM (Hill et al., 2021).

The EBM defense uses a classifier trained with labelled natural images and an EBM trained with unlabelled natural images. The two networks are trained independently, which is a key advantage of

EBM defense over adversarial training and randomized smoothing. Since the EBM is independent of the classifier, EBM defense has the potential to secure many classifiers across diverse tasks with a single defensive model, while existing methods are typically tailored to a single method. Starting with a naturally classifier trained on natural images $f(x)$, we define its robust counterpart as

$$F(x) = E_{T(x)}[f(T(x))] \tag{6}$$

where $T(x)$ is a random variable representing $K$ steps of the Langevin transformation (5) initialized from a state $x$. We cannot evaluate $F(x)$ directly so we approximate it using

$$\hat{F}_H(x) = \frac{1}{H} \sum_{h=1}^{H} f(\hat{x}_h) \quad \text{where} \quad \hat{x}_h \sim T(x) \text{ i.i.d.,} \tag{7}$$

where $f(.)$ is a forward pass of our classifier to return logits and where the accuracy of approximation is driven by the number of replicates $H$. Meaningful evaluation of adversarial defenses must be based on adaptive attack methods which are aware of both $f(x)$ and $T(x)$. For our attack we use the BPDA+EOT formulation from (Athalye et al., 2018; Tramer et al., 2020) to obtain the attack gradient

$$\Delta_{\text{BPDA+EOT}}(x, y) = \frac{1}{H_{\text{adv}}} \sum_{h=1}^{H_{\text{adv}}} \nabla_{\hat{x}_h} L\left(\frac{1}{H_{\text{adv}}} \sum_{h=1}^{H_{\text{adv}}} f(\hat{x}_h), y\right), \quad \hat{x}_h \sim T(x) \text{ i.i.d.} \tag{8}$$

which we use in the standard PGD framework to generate adversarial examples. Algorithm 2 gives a sketch of the defense evaluation.

---

**Algorithm 2** EBM Defense Algorithm

---

**Require:** Natural images $\{x_m^+\}_{m=1}^{M}$, EBM $U(x; \theta)$, classifier $f$, Langevin noise $\eta$, attack replicates $H_{\text{adv}}$, defense replicates $H_{\text{def}}$, $l_\infty$ radius $\varepsilon$, attack step size, $\alpha$, Langevin steps $K$
**Ensure:** Defense record $\{D_m\}_{m=1}^{M}$ for each image initialized as ones.
    **for** $1 \leq i \leq M$ **do**
        select$(X_i, y_i)$from batch
        Randomly initialize adversary $\hat{X}_0$ inside $L_\infty$ ball around $X_i$
        **for** $1 \leq j \leq N$ **do**
            $c_j = \arg\max_\ell [\hat{F}_{H_{\text{adv}}}(\hat{X}_{j-1})]_\ell$
            $\Delta_j = \Delta_{\text{BPDA+EOT}}(\hat{X}_{j-1}, y_i)$
            **if** $c_j \neq y_i$ **then**
                $c_j' = \arg\max_\ell [\hat{F}_{H_{\text{def}}}(\hat{X}_{j-1})]_\ell$
                **if** $c_j' \neq y_i$ **then**
                    $D_i = 0$
                **end if**
            **end if**
            $\hat{X}_j = \text{PGD}(\hat{X}_{j-1}, \Delta_j, \varepsilon, \alpha)$
        **end for**
    **end for**

---

# F   EBM DEFENSE EXPERIMENT DETAILS AND DIAGNOSTICS

Table 5: Defense for $l_\infty$ against high-power whitebox attacks on ImageNet and on CIFAR-10.

| Dataset | Nat | Adv | $H_{\text{adv}}$ | $H_{\text{def}}$ | samples |
|---------|-----|-----|------|------|---------|
| ImageNet | 0.683 | 0.38 | 32 | 64 | 1600 |
| CIFAR-10 | 0.858 | 0.64 | 48 | 128 | 5120 |

To verify the integrity of our results we ran an attack with heavily increased resources for ImageNet and CIFAR-10 compared to our standard evaluation. While using these resources for all attacks is infeasible in practice, we want to ensure our defense maintains robustness as attacker resources

---
**Algorithm 3** Training a midrun sampler for EBM defense

---
**Require:** Natural images $\{x_m^+\}_{m=1}^M$, EBM $U(x;\theta)$, frozen pre-trained generator $g(z)$, Langevin noise $\eta$, Langevin steps $K$, EBM optimizer $h_U$, initial weights $\theta_0$, rejuvenation probability $p$, number of training iterations $T$.
**Ensure:** Weights $\theta_T$ for defensive EBM.
    Initialize image bank $\{X_i^-\}_{i=1}^N$ from generator using $X_i^- = g(Z)$
    **for** $1 \leq t \leq T$ **do**
        Select batch $\{\tilde{X}_b^+\}_{b=1}^B$ from data samples $\{x_m^+\}_{m=1}^M$.
        Get negative sample batch $\{\tilde{X}_{b,0}^-\}_{b=1}^B$ from $\{X_i^-\}_{i=1}^N$.
        Update $\{\tilde{X}_{b,0}^-\}_{b=1}^B$ with $K$ Langevin steps (5) to obtain negative samples $\{\tilde{X}_b^-\}_{b=1}^B$.
        Get learning gradient $\Delta_U^{(t)}$ using (4) with samples $\{\tilde{X}_b^+\}_{b=1}^B$ and $\{\tilde{X}_b^-\}_{b=1}^B$.
        Update $\theta_t$ using gradient $\Delta_U^{(t)}$ and optimizer $h_U$.
        Rejuvenate each $\tilde{X}_b$ from a pretrained generator $g$ with probability $p$.
        Return $\{\tilde{X}_b^-\}_{b=1}^B$ to $\{X_i^-\}_{i=1}^N$ by overwriting previous states.
    **end for**

---

increase. As shown in Table 5, our benchmark accuracy remains consistent when we increase the number of attack steps (from 50 to 200) and EOT attack replicates ($H_{\text{adv}}$).

We also demonstrate results over varying numbers of langevin steps $K$. We can see in Fig 6 that our sampling trajectory for defending imagenet at $K = 200$ is reasonable to achieve high natural image classification as well as robustness. On CIFAR-10 we are able to reduce the number of langevin steps from 1500 in Hill et al. (2021) to $K = 500$, which greatly reduces our compute overhead and demonstrates that a non-convergent model can outperform a convergent model ($+13\%$ robustness) while using less steps.

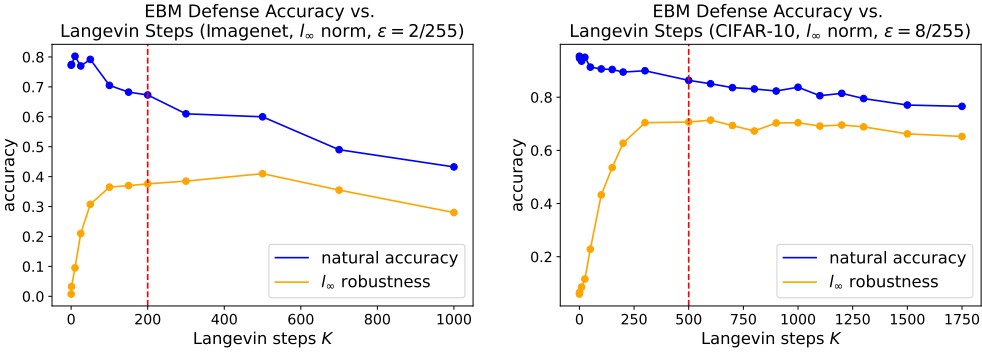

Figure 6: Accuracy over varying numbers of langevin steps $K$ for both the CIFAR-10 and Imagenet experiments. For CIFAR-10 we used $H_{\text{adv}} = 24$ in these experiments to compare against Hill et al. (2021) so the accuracy reported here is slightly higher than in table 2 that uses $H_{\text{adv}} = 48$

## G    Importance of Learning Rate Annealing

This section demonstrates the importance of learning rate annealing for learning a robust energy landscape. We repeat the midrun and longrun learning experiments for CIFAR-10 except that we never anneal the learning rate. We then sample with the models for 1500 steps for the model trained with the midrun method and 100K steps for the model trained with the longrun method. The results in Figure 7 show that learning rate annealing is essential for stabilizing both midrun and longrun trajectories.

The importance of annealing can be understood as follows. If the EBM is being updated with a very low learning rate, then samples from recent EBM snapshots can function as samples from the current EBM. In the case of midrun trajectory, annealing allows the model to robustify trajectories that are approximately as long as the lifetime of a persistent sample between rejuvenation. In the case of

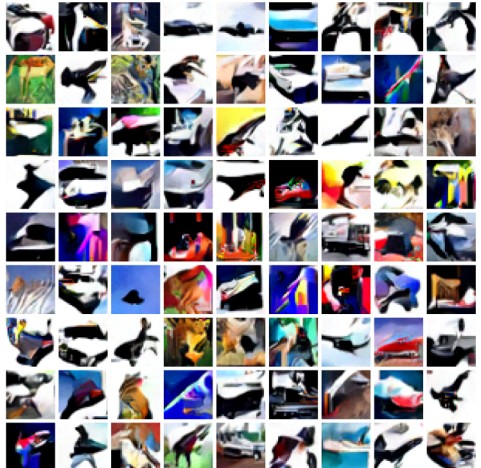 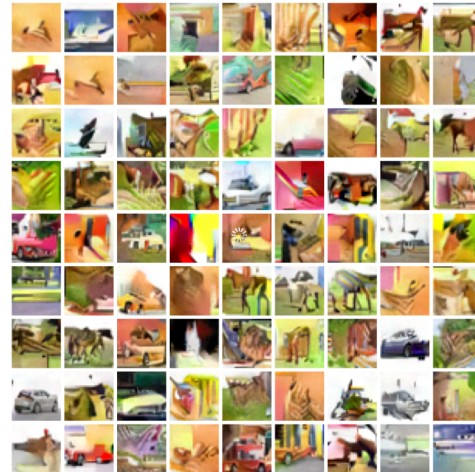

Figure 7: Ablation study showing the importance of annealing. *Left:* Samples from a non-annealed model trained with the midrun method after 1500 MCMC steps. *Right:* Samples from a non-annealed model trained with the longrun method ater 100K MCMC steps. MCMC samples were initialized from data. This shows that rejuvenation of the midrun trajectories from data and the separation of longrun samples into burn-in and update banks alone is not enough. Annealing ensures that samples from past EBMs function as approximate samples from the current EBM, since the weights are changing very slowly.

longrun learning, annealing allows the burnin samples to approximately reach the model steady-state before they are included in the update bank. This allows the persistent samples in the update bank to function as approximate steady-state samples from the current EBM, leading to proper modeling of probability mass.

## H   PRIOR EBM FOR LONGRUN LEARNING

This section discusses the choice of training longrun EBMs by using a fixed shortrun EBM as a prior distribution. Intuitively, the prior EBM learns a reasonable but imperfect approximation of the energy landscape that we know will have energy basins which leak to low-energy states. Using a fixed misaligned prior EBM allows the EBM which is being actively updated to focus on learning landscape features which were not correctly learned by the prior EBM. In particular, our longrun allows the actively updated EBM to focus on sealing the leaky energy basins of the prior EBM so that the full model learn the correct distribution of probability mass. By separating the burn-in and update banks and longrun learning, we guarantee that the prior EBM alone would lead to extreme oversaturation by the time a state reaches the update bank unless the actively updated EBM is counteracting the oversaturation tendency to preserve realism. This additional effect of directly driving states towards the oversaturated regions in the absence of a correcting force leads to improved stability for longrun models learned with a prior EBM compared to models learned without a prior EBM. The technique has precedent in ResNets. In both cases, it is easier to learn an approximate direct model first and then to approximate the residual that remains.

## I   COMPARISON TO SCORE-BASED DENOISERS, NORMALIZING FLOWS, AND DIFFUSION MODELS

The score-based model from (Song & Ermon, 2020) and its annealed Langevin dynamics process has recently been used for purifying adversarial signals (Lee & Oh, 2021; Yoon et al., 2021). One approach is add noise and the using the score model to denoise (Lee & Oh, 2021). The robustness of this method is upper-bounded by standard randomized smoothing (Cohen et al., 2019). Another approach is to initiate the Langevin process of the score model from a natural image as done in the EBM defense. A score model can be used in a langevin process $T(x)$ that is a direct analogue to the

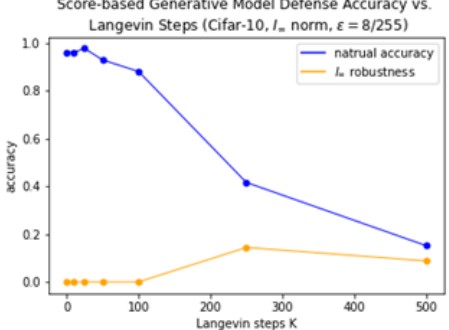 

Figure 8: Score-based Langevin experiment on the CIFAR-10 dataset. Left: Accuracy of natural and adversarial images resulting from a BPDA+EOT defense using a score-based model with an annealed langevin purification method for 125 samples over varying steps. Right: Samples received from this annealed langevin diffusion process over the same sampling lengths.

EBM langevin process in (5) and (8). Given a score model $S_\theta(x)$ for a low noise value $\sigma$, one can define the Langevin equation

$$X_t = X_{t-1} \frac{\eta^2}{2} S_\theta(X_{t-1}) + \eta Z_t$$

where $S_\theta(x) \approx \nabla_x \log q(x)$ since $\sigma$ is low. In both cases, the score-based model is trained with the loss:

$$\mathcal{L}(\theta) = \frac{1}{2L} \sum_{i=1}^{L} E_{q(x)} E_{q_{\sigma_i}(\tilde{x}|x)} \left[ \left\| \sigma_i s_\theta(\tilde{x}, \sigma_i) + \frac{\tilde{x} - x}{\sigma_i} \right\|_2^2 \right]$$

where $q_{\sigma_i}(\tilde{x}|x)$ represents adding Gaussian noise of standard deviation $\sigma_i$ to an image $x$ sampled from $q$.

We experimented with this process as a defense mechanism by selecting the smallest trained noise value $\sigma = 0.01$ and using the Langevin process as a method to purify adversaries. We evaluated this method using our BPDA+EOT attack framework over different numbers of Langevin steps during purification. In contrast to reports from Yoon et al. (2021), we were unable to obtain any significant defense using a pretrained score model when initializing sampling directly from adversarial or natural images. In Figure 8, one can see that the score-based model drives natural images toward saturation quickly, leading to a sharp decrease in natural classification that undermines the possibility of robustness from sampling. Since the score-based model does not perform sampling during training, and one cannot adjust the stability of its sampling process as we do in this work. While it is not immediately clear how to overcome this problem, we believe that defense with a score model is possible and we hope that our observation lead to efforts to stabilize the sampling paths of score models as we do for EBMs in this work.

To underscore our claims about the difficulty of calibrating the probability mass of a density model, we further investigate longrun MCMC samples from a normalizing flow and diffusion model. We find that the normalizing flow from the GLOW model (Kingma & Dhariwal, 2018) and the recovery likelihood diffusion model (Gao et al., 2020b) have misaligned steady-states as well. This shows that the problem of improper density estimation extends well beyond the EBM. Tractable density modeling with a normalizing flow does not prevent steady-state misalignment. These experiments corroborate our claim that log likelihood experiments in previous works are not able to detect the misaligned probability mass of many prior models. We believe that the calibration of the model steady-state is currently best diagnosed with longrun MCMC sampling because the distribution of longrun MCMC samples represents the probability mass of the model.

Figure 9 (*left*) displays initial and final states from a GLOW model density after 100K sampling steps. Despite the fact that the GLOW model has a fully tractable density, it is unable to learn a valid distribution of probability mass. Figure 9 (*right*) shows initial and final samples from the Recovery Likelihood T6 model after 100K steps of the conditional model at the lowest noise value. We observe the same oversaturation for the conditional density as for the unconditional density of a standard

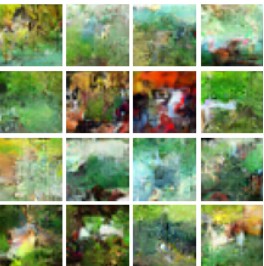 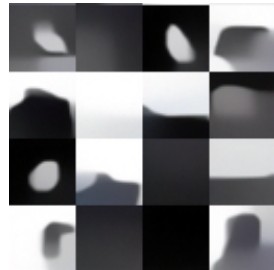

Figure 9: *Left:* MCMC samples after 100K steps using a GLOW model (Kingma & Dhariwal, 2018) trained on CIFAR-10. *Right:* MCMC samples after 100K steps using a conditional recovery likelihood model (Gao et al., 2020b) trained on CIFAR-10. MCMC samples were initialized from data samples. Neither model can correctly approximate the distribution of probability mass for the data density. The problem of steady-state misalignment extends beyond EBMs to many other generative density models. We tried several different temperatures close to 1 for the GLOW model and found equivalent results.

EBM. Code for the T1K model that the authors evaluate in their longrun experiments is not released so we have not yet been able to directly test their results. The T1K model is equivalent to an EBM version of the score model in Figure 8 which we have shown has a misaligned steady-state. Further, we note the longrun experiments with the T1K model are very misleading because the experiments use 100 steps with 1000 distinct conditional models and claim this is a longrun evaluation of 100K steps. The correct evaluation is to use 100K steps on a single conditional model. We strongly believe that the Recovery Likelihood model as originally presented has a misaligned steady-state like many other methods. We hope that the observations in our work can lead to efforts to stabilize the sampling trajectories of many existing models.

## J    REGARDING DENSITY ESTIMATION AND LOG LIKELIHOOD

In this work, we are primarily interested in learning density models that assign the majority of probability mass in realistic regions of the image space. This goal is, surprisingly, distinct from the goal of likelihood maximization. In particular, one can achieve high likelihood with mixture models where only an infinitesimal portion of mass is assigned to a mixture component that approximates the true density, while the majority of probability mass is assigned to a degenerate distribution (see Theis et al. (2016), "Great Log Likelihood but Poor Samples"). The steady-state of MCMC sampling with such a mixture distribution would concentrate on the degenerate distribution even though the structures of the true density exist in high-energy regions. Therefore, log likelihood cannot detect if a model has assigned probability mass in realistic region of the image space. Recent observations have shown that this situation is not just hypothetical (Nijkamp et al., 2020). Non-convergent EBMs are practical examples, since these models consist of a mixture of partially formed high-energy energy basins enabling effective shortrun synthesis in realistic regions of the image space and much lower-energy basins in unrealistic image regions that dominate the probability mass. The same misaligned landscape structure extends beyond EBM to RBMs (Decelle et al., 2021), score models, normalizing flows, and diffusion models (see Appendix I). Log likelihood can only indicate the presence of energy landscape features that are similar to the ground truth landscape, but it cannot detect whether these features will leak into lower-energy basins that represent the true mass distribution. Like shortrun sample quality, high log likelihood is often misleading false evidence that a model density concentrates on realistic images.

Overall, we believe that FID using true samples from the EBM and data samples is an appropriate, if rough, measure of successful density modeling. The subtlety involves generating true samples from the EBM. We believe that longrun MCMC sampling is the most practical and principled tool for obtaining approximate samples from the EBM steady-state, and our investigations show that density modeling with properly calibrated probability mass is currently most effectively accomplished using MCMC-based EBM training. Though we use the FID metric, our density modeling experiments are best described as realism preservation experiments rather than image generation experiments.

# K  LONGRUN LEARNING ALGORITHM

---

**Algorithm 4** Training a longrun sampler for density estimation

---

**Require:** Natural images $\{x_m^+\}_{m=1}^M$, EBM $U(x; \theta)$, frozen pre-trained generator $g(z)$ Langevin noise $\eta$, Langevin steps $K$, EBM optimizer $h_U$, initial weights $\theta_0$, update threshold $D$, number of training iterations $T$.

**Ensure:** Weights $\theta_T$ for EBM with stable longrun samples.

  Initialize burn-in image bank $\{X_i^*\}_{i=1}^{N_1}$ and update image bank $\{X_i^+\}_{i=1}^{N_2}$ from generator using $X_i^- = g(Z)$.

  Initialize update counts $\{d_i\}_{i=1}^{N_1}$ using $d_i \sim \text{Unif}(\{0, \ldots, D\})$.

  **for** $1 \le t \le T$ **do**

    Select batch $\{\tilde{X}_b^+\}_{b=1}^B$ from data samples $\{x_m^+\}_{m=1}^M$.

    Get initial sample batch $\{\tilde{X}_{b,0}^*\}_{b=1}^B$ from $\{X_i^*\}_{i=1}^{N_1}$ and $\{\tilde{X}_{b,0}^-\}_{b=1}^B$ from $\{X_i^-\}_{i=1}^{N_2}$.

    Get counts $\tilde{d}_b$ corresponding to samples $\{\tilde{X}_{b,0}^*\}_{b=1}^B$.

    Update $\{\tilde{X}_{b,0}^-\}_{b=1}^B$ with $K$ Langevin steps (5) to obtain negative samples $\{\tilde{X}_b^-\}_{b=1}^B$.

    Update $\{\tilde{X}_{b,0}^*\}_{b=1}^B$ with $K$ Langevin steps (5) to obtain updated burn-in samples $\{\tilde{X}_b^*\}_{b=1}^B$.

    Get learning gradient $\Delta_U^{(t)}$ using (4) with samples $\{\tilde{X}_b^+\}_{b=1}^B$ and $\{\tilde{X}_b^-\}_{b=1}^B$.

    Update $\theta_t$ using gradient $\Delta_U^{(t)}$ and optimizer $h_U$.

    Update the burn-in count $\tilde{d}_b \leftarrow \tilde{d}_b + 1$.

    **for** $1 \le b \le B$ **do**

      **if** $\tilde{d}_b \ge D$ **then**

        Randomly overwrite one $\tilde{X}_b^-$ using $\tilde{X}_b^*$.

        Rejuvenate $\tilde{X}_b$ from generator and set $\tilde{d}_b$ to 0.

      **end if**

    **end for**

    Return $\{\tilde{X}_{b,0}^*\}_{b=1}^B$ from $\{X_i^*\}_{i=1}^{N_1}$ and $\{\tilde{X}_b^-\}_{b=1}^B$ to $\{X_i^-\}_{i=1}^{N_2}$ by overwriting previous states.

  **end for**

---

## L    SHORTRUN SYNTHESIS RESULTS

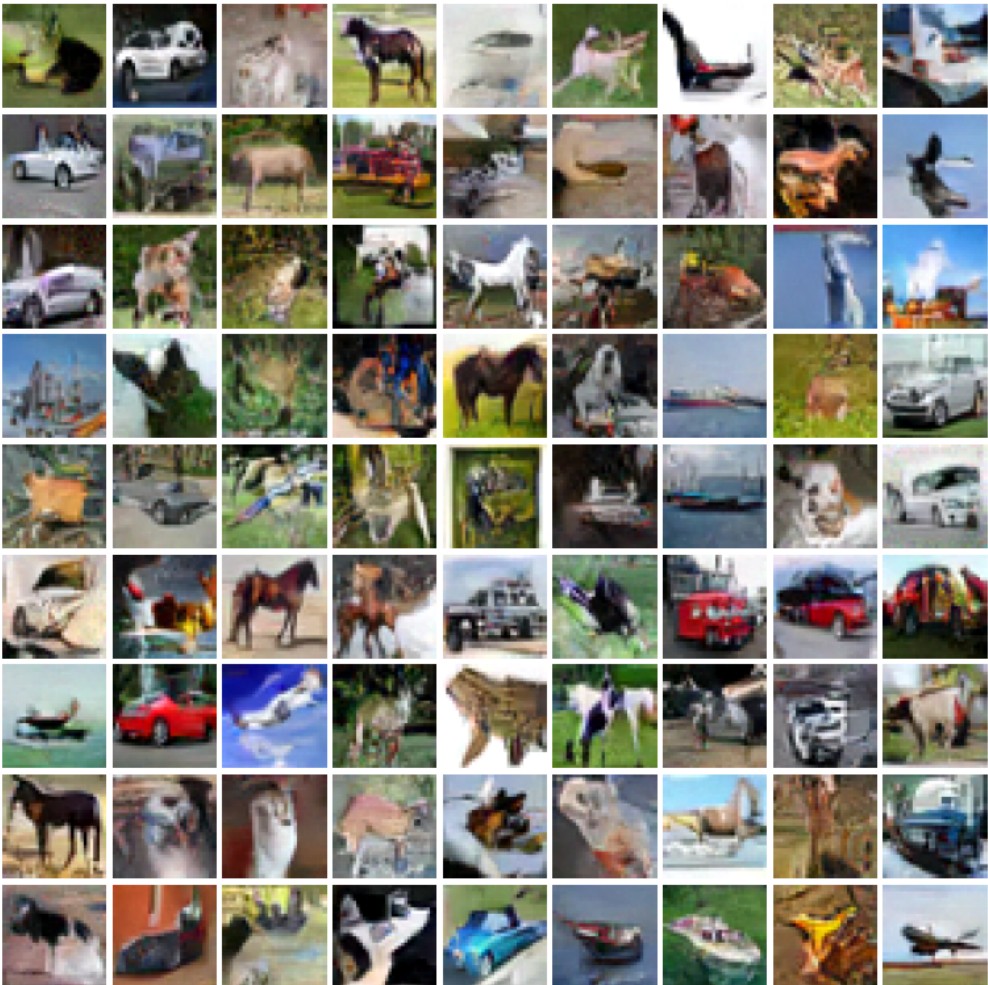

Figure 10: Shortrun samples from CIFAR-10 EBM at resolution $32 \times 32$.

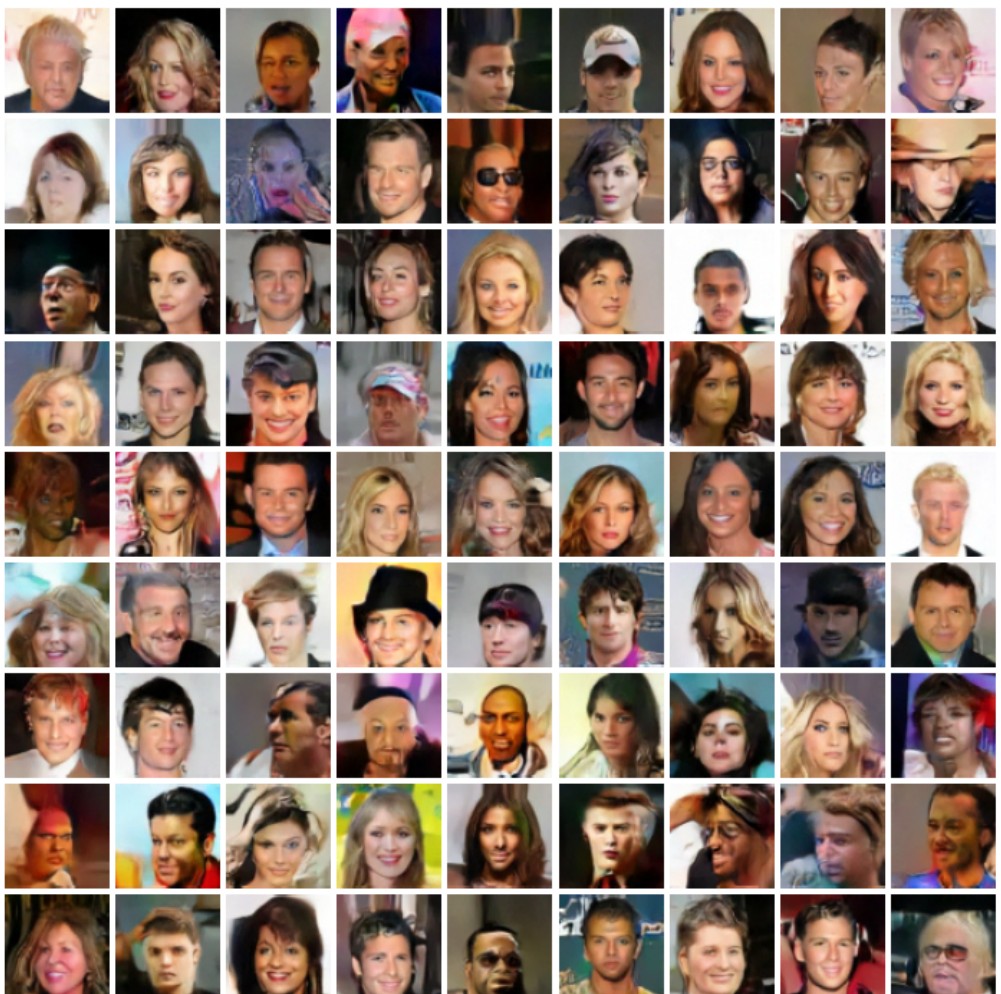

Figure 11: Shortrun samples from Celeb-A EBM at resolution $64 \times 64$.

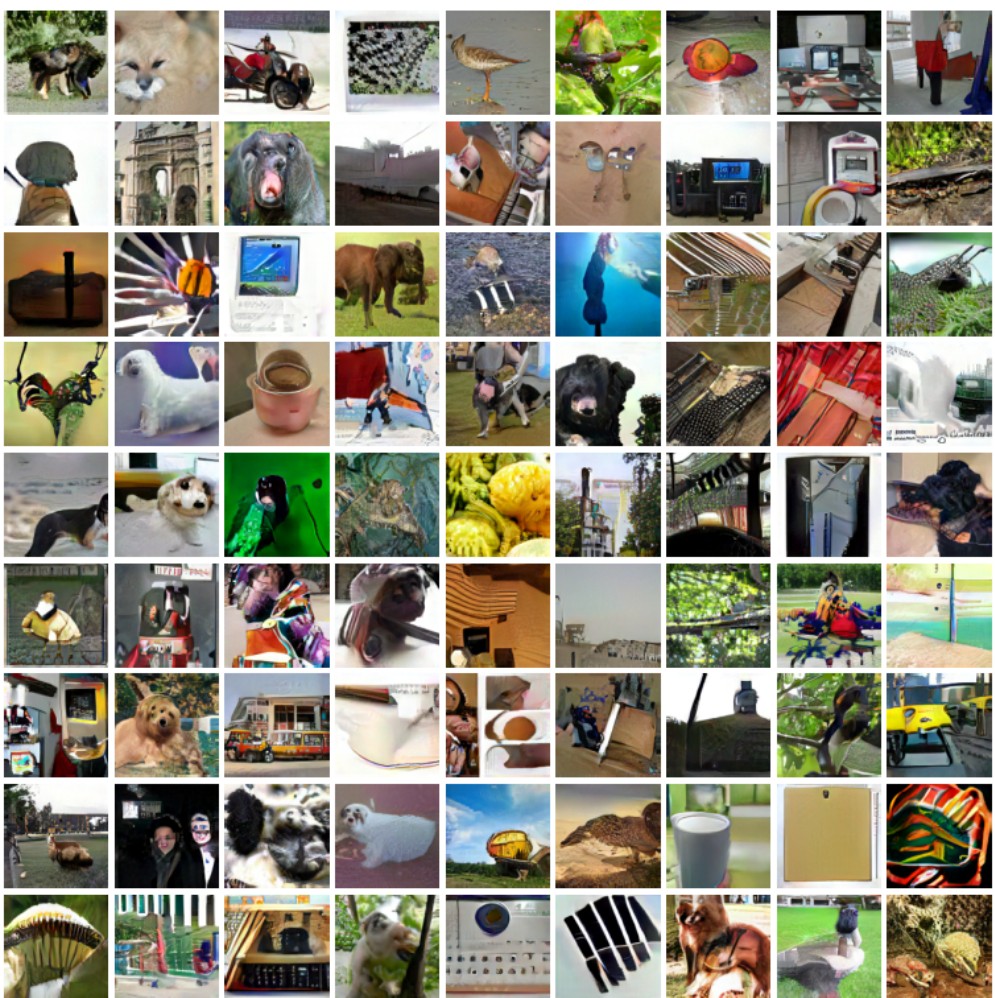

Figure 12: Shortrun samples from ImageNet EBM at resolution $128 \times 128$.

# M    Longrun Synthesis Results

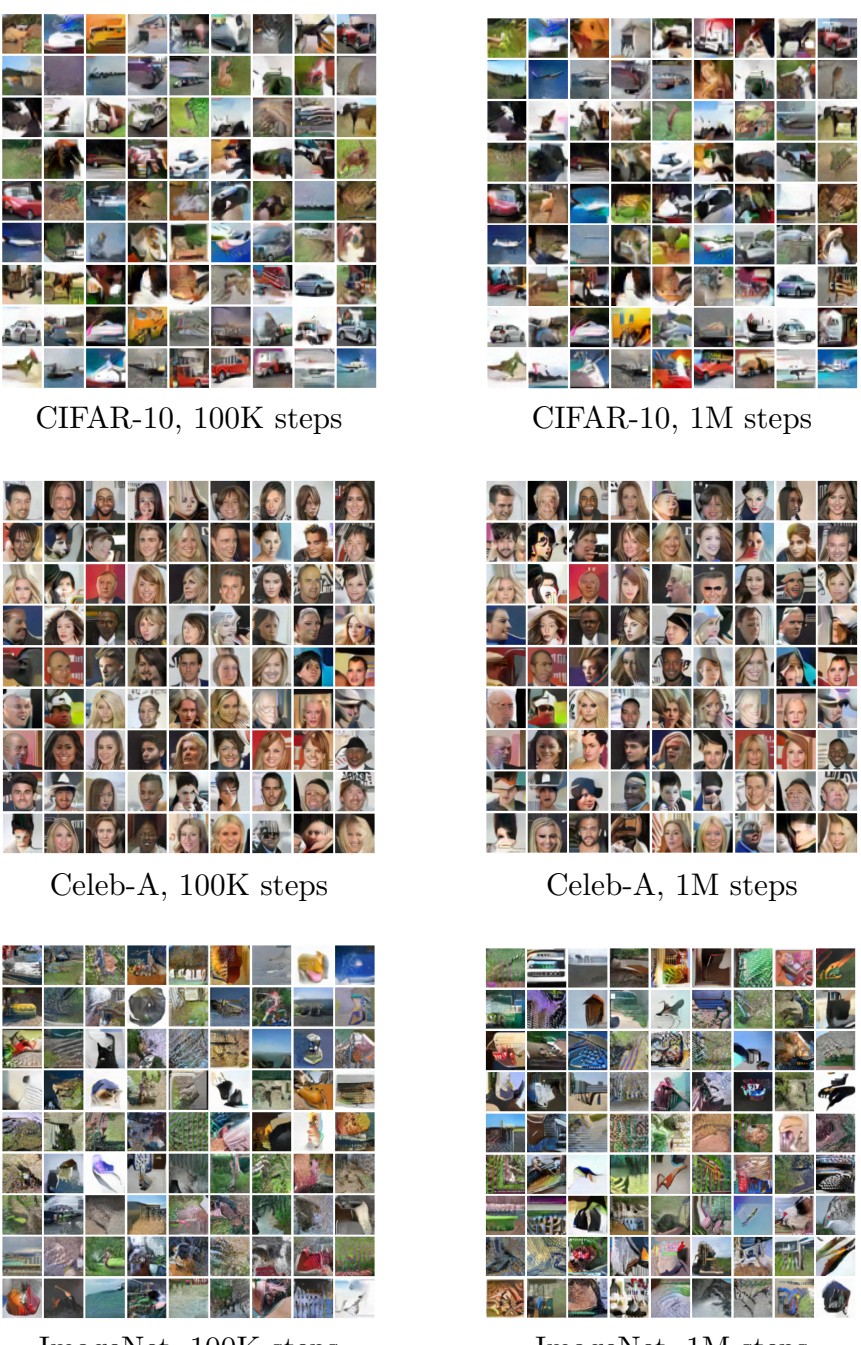

CIFAR-10, 100K steps          CIFAR-10, 1M steps

Celeb-A, 100K steps          Celeb-A, 1M steps

ImageNet, 100K steps          ImageNet, 1M steps

Figure 13: Longrun samples at 100,000 steps and extremely longrun samples at 1 million steps for EBMs trained on three datasets. Our initialization is able to preserve a high degree of realism over the first 100K steps, and a reasonable degree of realism over very long trajectories. While oversaturation and distortion is noticeable for some samples using 1M steps, many samples have reasonable appearance and there is high diversity. Our method makes significant progress towards aligning longrun samples with high-quality samples from training to ensure that the model is a valid density.

# N    TABLES OF EXPERIMENT PARAMETERS

This section gives a full list of experiment parameters. Some notable minor additions beyond what are discussed in the text are a fixed MCMC temperature parameter to regulate gradient strength at the beginning of training, a gradient clipping parameter to restrict the magnitude of EBM and generator weight updates. We only use gradient clipping for shortrun experiments.

We use the annealing schedule

$$\gamma_{\text{anneal}} = [(10^{-4}, 0), (10^{-5}, 50000), (10^{-6}, 75000), (10^{-7}, 100000), (10^{-8}, 125000)]$$

in all experiments that involve annealing, where the pairs denote a learning rate and model update step at which that learning rate is first used. We denote this schedule as $\gamma_{\text{anneal}}$ in the tables.

| Shortrun Training | | | |
|---|---|---|---|
| Dataset | Celeb-A | CIFAR-10 | ImageNet |
| Training Steps | 150000 | 100000 | 300000 |
| Data Epsilon | 1e-2 | 1e-2 | 1e-2 |
| EBM LR | 1e-4 | 1e-4 | 1e-4 |
| EBM Optimizer | Adam | Adam | Adam |
| EBM Gradient Clip | 0 | 0 | 50 |
| Langevin Epsilon | 3e-3 | 5e-3 | 3e-3 |
| MCMC Steps | 75 | 100 | 50 |
| Rejuvenation Probability | 0.5 | 0.5 | 0.5 |
| MCMC Temperature | 1e-6 | 1e-4 | 1e-7 |
| Max Update Rounds | 2 | 2 | 2 |
| Persistent Bank Size | 10000 | 10000 | 10000 |
| Generator LR | 1e-4 | 1e-4 | 5e-5 |
| Generator Optimizer | Adam | Adam | Adam |
| Generator Gradient Clip | 0 | 0 | 50 |
| Generator Batch Norm | No | Yes | No |

| Shortrun Evaluation | | | |
|---|---|---|---|
| Dataset | Celeb-A | CIFAR-10 | ImageNet |
| Langevin Epsilon | 3e-3 | 5e-3 | 3e-3 |
| MCMC Steps | 300 | 350 | 320 |
| MCMC Temperature | 1e-6 | 1e-4 | 1e-7 |

**Defense Training**

| Dataset | CIFAR-10 | ImageNet |
|---|---|---|
| Training Steps | 150000 | 150000 |
| Data Epsilon | 2.5e-2 | 3e-2 |
| EBM LR | $\gamma_{\text{anneal}}$ | $\gamma_{\text{anneal}}$ |
| EBM Optimizer | Adam | Adam |
| Langevin Epsilon | 1.25e-2 | 2e-2 |
| MCMC Steps | 100 | 100 |
| Rejuvenation Probability | 0.05 | 0.05 |
| MCMC Temperature | 1e-4 | 1e-5 |
| Persistent Bank Size | 25000 | 10000 |

**Defense Evaluation**

| Dataset | CIFAR-10 | ImageNet |
|---|---|---|
| Adversarial Steps | 50 | 50 |
| Adversarial Epsilon | $\frac{8}{255}$ | $\frac{2}{255}$ |
| Adversarial Eta | $\frac{2}{255}$ | $\frac{1}{255}$ |
| EOT Attack Reps | 48 | 16 |
| EOT Defense Reps | 128 | 64 |
| Langevin Steps | 500 | 200 |
| Langevin Epsilon | 1.25e-2 | 2e-2 |
| MCMC Temperature | 1e-4 | 1e-5 |

**Longrun Training**

| Dataset | Celeb-A | CIFAR-10 | ImageNet |
|---|---|---|---|
| Training Steps | 250000 | 250000 | 250000 |
| Data Epsilon | 2e-2 | 2e-2 | 2e-2 |
| EBM LR | $\gamma_{\text{anneal}}$ | $\gamma_{\text{anneal}}$ | $\gamma_{\text{anneal}}$ |
| EBM Optimizer | Adam | Adam | Adam |
| Langevin Epsilon | 1e-2 | 1e-2 | 1e-2 |
| MCMC Steps | 100 | 100 | 100 |
| Burn-in Update Rounds | 750 | 750 | 750 |
| MCMC Steps Burn-in | 100 | 100 | 100 |
| MCMC Temperature | 1e-4 | 1e-4 | 1e-5 |
| Tau | 1.5e-1 | 1.5e-1 | 1.5e-1 |
| Persistent Bank Size | 15000 | 10000 | 15000 |
| Burn-in Bank Size | 1000 | 1000 | 1000 |

**Prior EBM**

| Dataset | Celeb-A | CIFAR-10 | ImageNet |
|---|---|---|---|
| Training Steps | 150000 | 150000 | 150000 |
| Data Epsilon | 2e-2 | 2e-2 | 1.5e-2 |
| EBM LR | 1e-4 | 1e-4 | 1e-4 |
| EBM Optimizer | Adam | Adam | Adam |
| Langevin Epsilon | 1e-2 | 1e-2 | 1e-2 |
| MCMC Steps | 100 | 50 | 100 |
| Rejuvenation Probability | 0.2 | 0.2 | 0.2 |
| MCMC Temperature | 1e-4 | 1e-4 | 1e-5 |
| Tau | 1.5e-1 | 1.5e-1 | 1.5e-1 |
| Persistent Bank Size | 10000 | 10000 | 10000 |

