# OpenReview forum: "EBM Life Cycle: MCMC Strategies for Synthesis, Defense, and Density Modeling"
_ICLR.cc/2022/Conference — ICLR 2022 Submitted_

### Official Review · Reviewer_oMaP · 2021-10-28

**Correctness:** 3
**Technical Novelty And Significance:** 1
**Empirical Novelty And Significance:** 2
**Recommendation:** 5
**Confidence:** 4

**Main Review:**

The paper is well written and easy to follow. The empirical performance looks good compared to the baseline models (although the fairness requires further investigation).
The weakness of the paper is that the contribution seems to be incremental, as most of the techniques used in this paper have been proposed somewhere else. Below are some of my concerns.

1. For image generation, why not use a pretrained generator? And if the generator already provide high quality samples (e.g., SNGAN), what is the purpose of using EBM after it? Did the baseline EBMs use generator initialization? Also, I did not see a comparison to Xie et al. (2018).

2. How does the pretained generator initialization compared with data samples? How important if annealing? Is annealing applied to other baseline methods? Lack Ablation studies here.

3. Where is the density estimation result for longrun EBM?

**Summary Of The Paper:**

This work presents different MCMC initialization techniques for training EBM with differerent lengths, for the purposes of image generation, adversarial defense, and density estimation respectively. More specifically, for shortrun image generation, the author proposed a hybrid persistent cooperative initialization that mitigate the lack of diversity issue of EBM learning with generator initialization. For midrun adversarial defense, the author proposed a pretrained generator rejuvenation to scale up EBM defense. For longrun density estimation, the author proposed rejuvenation methods and regularization trick to correct oversaturation.

**Summary Of The Review:**

Overall, the paper provide some interesting empirical findings for EBMs with different MCMC lengths. However, current experiment results and lack of novelty make it a bit under the bar of ICLR.

---

> ### Author Response · Authors · 2021-11-20
> **Response to Reviewer oMaP**
>
> We appreciate your time and insightful comments. Our responses to your review are below.
>
> * *"For image generation, why not use a pretrained generator?"*: It is widespread practice to investigate EBMs primarily for their ability to generate samples, and we devote one section of the paper to this common goal. In the majority EBM works that investigate synthesis, samples are not initialized from a pretrained model. If generator models are used, they are nearly always learned concurrently with the EBM.
>
> * *"I did not see a comparison to Xie et al. (2018)."*: The original publication of Xie et al. (2018) did not report FID on CIFAR-10. We have since found that the scores can be found in a separate submission [a]. The FID score for CIFAR-10 is 33. Our method yields significant improvement. We will include this in future versions.
>
> * *"Lack Ablation studies"*: We agree that including these ablation studies will improve our work. We have included ablative studies for the learning rate annealing in the Supplementary appendix and will include this in future revisions. We also are performing ablative studies with data initialization and will include those results in the later revisions.
>
> * *"Where is the density estimation result for longrun EBM?":
> Our density estimation results are the longrun FID scores in Table 4 and visualizations of longrun samples in Figure 11. Please refer to the general statement for our explanation of why we believe these are an appropriate evaluation to investigate density modeling.
>
> * *"current experiment results and lack of novelty"*: Please refer to our general response regarding novelty.
>
> [a] http://www.stat.ucla.edu/~ywu/PAMI.pdf

---

### Official Review · Reviewer_PEpN · 2021-10-29

**Correctness:** 3
**Technical Novelty And Significance:** 2
**Empirical Novelty And Significance:** 2
**Recommendation:** 3
**Confidence:** 4

**Main Review:**

Strengths:

This paper addresses a large issue in the EBM space. Current methods for training these models are difficult to tune, unstable, and slow compared to other generative modeling approaches. The proposed fix combining ideas from CoopNets and PCD may provide a best-of-both-worlds approach to training EBMs while adding no overhead on top of either of these training methods.

Further, this paper correctly brings attention to many real issues in the EBM space such as the sample quality / density model quality trade-off. As well, the paper addresses adversarial robustness which appears to be a powerful and promising of EBMs. While I am not an expert in adversarial robustness (and cannot make super-strong claims about the correctness of these experiments), it appears to me that the robustness results give a solid improvement over [6] and many other techniques purely meant for this task.


Weaknesses:

While the authors provide some high-level intuition for their proposed modifications to CoopNets/PCD, they do not provide any theoretical justification for their proposed approach. Have you considered other ways to improve the diversity of the CoopNet samples? Could this be due to the approximations that are made with the standard CoopNet learning algorithm (which the authors mention)? You could regularize entropy as in [5, 4]. I feel the paper would be made much stronger if more care was placed on the details and more theoretical justification was given.

The experimental details in the paper are quite scant. The authors do not mention anything about the optimizers used, the learning rates, the batch sizes, or training training time. As such, the results in this paper are not reproducible. I am aware the authors have included code, but these details should be accessible from the paper (at least in the appendix). Further, from an investigation of the code, there appear to be a number of additional experimental hyper-parameters (such as gradient clipping and energy tempering) which are not once mentioned in the paper. These modifications to the training objective can be very responsible for the success/failure of EBMs when training in this way and there has been a good deal of work discussing it.

This work is attempting to improve stability and success of EBM training. Many of these tricks (or hacks depending on your point of view) are there to accommodate for many of the issues that this paper proposes to address (such as high variance of negative sample gradients from training on newly resampled examples) so it should be made clear which, if any, of these tricks are no longer needed.

I have some issues with the evaluation used in the long-run sampling for density estimation section. The authors claim early in the paper that sample quality can be a misleading indicator of model quality (where quality is evaluated by maximum likelihood), which I completely agree with. I was then disappointed to find the only quantitative result in this action was FID which is a qualitative measure of sample quality. If the authors want to argue that their training procedure learns a better density model then there are many alternative evaluations which could be used -- and have been used in recent EBM work. You could use AIS/RAISE to estimate upper/lower-bounds on likelihood as in [1, 2]. Or, you could use your training procedure to train a tractable likelihood model and then evaluate using the model’s known likelihood as in [3, 4].


Minor: Typo in Table 2.

[1] Du, Yilun, and Igor Mordatch. "Implicit generation and generalization in energy-based models." arXiv preprint arXiv:1903.08689 (2019).
[2] Gao, Ruiqi, et al. "Learning energy-based models by diffusion recovery likelihood." arXiv preprint arXiv:2012.08125 (2020).
[3] Song, Yang, et al. "Sliced score matching: A scalable approach to density and score estimation." Uncertainty in Artificial Intelligence. PMLR, 2020.
[4] Grathwohl, Will, et al. "No MCMC for me: Amortized sampling for fast and stable training of energy-based models." arXiv preprint arXiv:2010.04230 (2020).
[5] Dieng, Adji B., et al. "Prescribed generative adversarial networks." arXiv preprint arXiv:1910.04302 (2019).
[6] Hill, Mitch, Jonathan Mitchell, and Song-Chun Zhu. "Stochastic security: Adversarial defense using long-run dynamics of energy-based models." arXiv preprint arXiv:2005.13525 (2020).

----------Post author response------------

I appreciate the authors responding to my feedback and thank them for the changes they have made to the paper. Unfortunately, I do not find their arguments convincing regarding density estimation and the use of FID to evaluate the method. As well, if the authors were able to demonstrate that their method simplifies EBM training while offering similar quality results, this would be compelling, but I do not feel this was done in the work. The proposed method is more complicated than CoopNets and PCD and provides additional hyper-parameters to tune. If the authors were able to make a compelling argument that their method allows us to learn a better density model, then I would support acceptance of this work, but I do not believe this was done. For this reason, I will keep my score the same. There are some interesting ideas presented in this work, but I do not feel the results presented demonstrate that they are a sufficient contribution to the field for acceptance.


**Summary Of The Paper:**

This paper discusses 3 applications for EBMs: image generation, adversarial robustness, and density modeling. The authors discuss how the standard methods for training EBMs do not adequately address the latter two of these applications. For this reason, they propose a new method which combines ideas from 2 popular EBM training methods; CoopNets, and PCD, to alleviate some issues with both methods. The authors present results on the 3 tasks mentioned and their method appears favorable compared to other EBM variants.

**Summary Of The Review:**

This paper presents a new method for training EBMs which combines features of two popular training approaches: CoopNets and PCD. While the proposed method makes sense, and appears to work, there are considerable issues with the method’s evaluation, experimental details, and theoretical justification. Thus, in its current form, I do not advocate for its acceptance.

---

> ### Author Response · Authors · 2021-11-20
> **Response to Reviewer PEpN [1/2]**
>
>
> We are glad to hear that you find the problems addressed by our work by be significant problems in the EBM space. One of the central contributions of our work is to disseminate observations that demystify practical outcomes for EBM training, which is widely regarded as especially difficult to use and unstable. Your responses will help us improve the quality of our work.
>
> * *While the authors provide some high-level intuition for their proposed modifications to CoopNets/PCD, they do not provide any theoretical justification for their proposed approach.*: While a full theoretical explanation is daunting, there are some practical considerations that explain the need for increased diversity in Cooperative Learning. If shortrun trajectories are used, it becomes difficult for MCMC samples to develop stable trajectories across the image space when generated from a low-diversity initialization. This happens because there is a upper limit on the maximum step size for a Langevin chain that is compatible with stable learning. This limit is determined by the geometry of the data distribution, and depends on the exact data scaling (e.g. pixel range [-1, 1]). On the other hand, simple geometry determines the per-step distance that the chain must travel towards a distant data mode from a low-diversity starting point with shortrun sampling. When the distance that the MCMC paths needs to travel to cover the data manifold forces the Langevin gradients to exceed the limit determined by data geometry, learning becomes unstable. This explanation motivated the development of our shortrun learning method.
>
>
> * *You could regularize entropy as in [5, 4]*: Our experiments show that entropy generated by MCMC sampling from diverse initialization produces higher quality and more scalable results than the variational methods in [5, 4]. While MCMC-based approaches and variational approaches can both be justified in theory, in practice the theoretical assumptions of both methods are assuredly not met. In particular, neither MCMC samples from the generator, nor direct samples from the generator for variational methods, are representative of true steady-state samples, as discussed in (Nijkamp et al. 2020). Therefore, the theoretical considerations in [5, 4] must be viewed in the lens of a heavily violated convergence assumption. In the image generation section, we embrace the violated assumption for image generation and focus on practical aspects considerations to design our initialization while remaining compatible with the theory behind cooperative learning.
>
> * *The experimental details in the paper are quite scant.*: We fully agree that including more experimental details directly in the paper is necessary. We will add complete details of all our training and testing hyperparameters to the appendix.
>
> * *there appear to be a number of additional experimental hyper-parameters (such as gradient clipping and energy tempering)*: Our learning implementations are quite minimal. Beyond what is discussed in the paper, we have a parameter for gradient clipping of the EBM and generator, a gradient clipping parameter for the Langevin chains, and a fixed temperature to adjust the scale of the energy function. The fixed temperature is only necessary to ensure that training is smoothly initialized by adjusting the magnitude of the Langevin gradients early in training to be approximately the magnitude gradients will converge to later in training. A similar technique is discussed in the note [a] that accompanies the code from (Nijkamp et al. 2020). This parameter merely stabilizes the beginning of training. It could be avoided by a suitable network weight initialization, but we find it is simpler to add a scalar temperature adjustment. This is equivalent to the decoupled step sizes for the gradient and noise term that are widely used in the EBM literature. Formulation as a temperature adjustment rather than decoupled step sizes is more consistent with true Langevin equation. Langevin gradient clipping is only used for shortrun experiments, where we expect the landscape to have unstable features. No gradient clipping is used for midrun or longrun experiments for either Langevin chains or network update gradients. The Langevin clipping threshold was set to a high value and clipping only occurs for samples that are on the verge of oversaturation. Since the reviews have been released, we retrained models without Langevin gradient clipping and got nearly equivalent scores (see Supplementary material). We will remove Langevin gradient clipping from future implementations.

---

> > ### Author Response · Authors · 2021-11-20
> > **Response to Reviewer PEpN [2/2]**
> >
> > * *which, if any, of these tricks are no longer needed*: We find that gradient clipping of the EBM and generator network gradients improves the stability of learning process, but **only for shortrun learning**, and not midrun and longrun learning. This is likely due to the effects of shortrun samples which begin to experience oversaturation before rejuvenation. Gradient clipping for the EBM and generator gradients is a widespread optimization tool, and not much more of a departure from standard Maximum Likelihood than using Adam. Beyond this, there are no hacks, just design choices for the MCMC implementation and rejuvenation process. Since the submission of the paper, we have re-verified each experiment presented herein and are confident that they could be reproduced by a third party when we publicly release the trained models and some minor code updates.
> >
> > * *issues with the evaluation used in the long-run sampling for density estimation section*: Please see our general response regarding log likelihood. We note that although [1,2,3,4] all claim to have performed proper density estimation and compute approximate log likelihoods, none of these works correctly diagnoses that the steady state of the learned model concentrates on unrealistic images. This further corroborates our claim that log likelihood cannot detect density miscalibration. The nonconvergence of [1] and variational methods such as [4] are known from (Nijkamp et al. 2020). Appendix G demonstrates that [3] does not provide proper density calibration of steady-state mass. There is an experiment in [2] that claims to evaluate longrun trajectories, but the implementation is very misleading because 100 steps from 1000 distinct conditional models are presented as a single trajectory of 100k steps. It is unclear why 100k steps with a single conditional model was not performed, since this would provide a much more reliable estimate of a conditional steady-state than using 100 steps. The model and training code for the exact longrun experiment in [2] are not released. In the supplementary appendix we show that a different version of the learning framework has oversaturated samples. Further, the authors of [2] note that method is equivalent to [3], which we show is misaligned in Appendix G. One of the key contributions of our work is bringing much-needed attention to the subtleties of calibrating the mass of a density model.
> >
> > [a] https://github.com/point0bar1/ebm-anatomy/blob/master/ebm-anatomy-appendix.pdf

---

### Official Review · Reviewer_8SkN · 2021-10-29

**Correctness:** 3
**Technical Novelty And Significance:** 2
**Empirical Novelty And Significance:** 2
**Recommendation:** 3
**Confidence:** 4

**Main Review:**

**Strengths**
The paper investigates training heuristics for energy-based models.

**Weaknesses**

I am not convinced by the argument that the performances are "state-of-the-art".
- In Table 1, the proposed EBM method clearly does not outperform GANs and recently proposed diffusion / score-based generative models.
- In Table 2, the number of *ours* has a typo of 0.0.566 in CIFAR10 and has worse natural accuracy than the EBM method in Hill et al., and ImageNet results are clearly not outperforming baselines (although the state-of-the-art claim avoids this).
- There are no density estimates on the test set, only FID, since the method does not discuss how to estimate the partition function.
- We have yet to discuss the amount of compute and time needed to perform training and inference with EBMs, which are quite slow as well.

The paper does not have many technical novelties. Most of the discussion is about the usage (or not) of persistent initialization and pretrained generator for "rejuvenation". The "rejuvenation model" is so important in all three cases, that a reasonable alternative is to use a good "rejuvenation model" and avoid EBM entirely. The heuristics themselves might be helpful in EBM learning, but it is questionable why we would adopt an EBM in the first place if it does not perform well on image generation/density estimation compared to other models (GANs, flow models, diffusion models).
- For example, a pretrained SNGAN is used to rejuvenate EBMs (section 4), yet the model itself outperforms EBMs in terms of FID (see Table 1).

If the method claims "density estimation", then we should expect to see results other than FID (since that is image generation), even if partition function is not possible to compute, it can still help to see if density estimation have other uses, such as out-of-distribution detection.

## Post rebuttal response
I appreciate the authors taking the time to discuss and address my concerns. Unfortunately, I will keep my scores as is.

- Similar to reviewer pepN, I am not entirely convinced about using FID in the "density modeling" section. Sure, density modeling does not require partition functions, and both FID and likelihood are flawed as measurements of image quality, but the argument used in this paper can also be used in GANs (it is also modeling some kind of density, and you can evaluate FID with it)? EBMs are more expressive than GANs in the sense that they also give un-normalized densities, but the paper never used these densities for any meaningful tasks (like out-of-distribution detection). The rebuttal also mentions that "(ours) represent a significant departure from mainstream approaches for learning and evaluating density models"; I don't see how evaluating FID is a significant departure, most EBM-based work after 2019 already report FIDs?
- Existing work has shown that diffusion models can beat GANs on ImageNet (even with higher resolutions): https://github.com/openai/guided-diffusion; this includes unconditional and conditional models.
- The "technically novel" bit in this paper seems to be using pretrained generators for rejuvenation, and if there are models that already beats EBM in image generation, why would we train and use another EBM instead? The rebuttal mentions that "Alternatives such as variational approximation yield much worse distributional approximations than MCMC", which might be true in principle, but can be quite far from what happens in practice (since MCMC can have slow mixing speed). Again, diffusion models are inspired by variational inference and seem to have quite good FIDs.

All in all, I find that the paper and rebuttal make some claims in favor of EBMs (and the paper itself) that I cannot fully agree with.


**Summary Of The Paper:**

The paper discusses learning strategies for energy-based models, short sampling for image generation, midrun sampling for adversarial defense, and longrun sampling for density estimation. The paper claims these methods achieve significant performance gain across the three applications and achieved state-of-the-art performances.

**Summary Of The Review:**

The paper spans various different problems, but makes relatively little contribution to each of them. The claims over empirical performances are inflated. The EBM-based methods do not have an advantage over score-based / diffusion generative models in terms of generation and likelihood evaluation (which also avoids estimating partition functions).

---

> ### Author Response · Authors · 2021-11-20
> **Response to Reviewer 8Skn**
>
>
> Thank you for raising important questions regarding our motivations and goals for learning EBMs. Your response will help us better explain our work in the context of other generative models. Responses to specific points are below.
>
> * *In Table 1, the proposed EBM method clearly does not outperform GANs and recently proposed diffusion / score-based generative models*: Our claim for state-of-the-art synthesis refers specifically to unnormalized image densities. Our method demonstrates superior results compared with an array of prior unconditional and conditional EBMs. While GANs achieve better FID scores on Celeb-A and CIFAR-10, our results on 128x128 ImageNet are better than GAN models that use a batch size of approximately 128. We believe that our method has the potential to match the SOTA FID scores for unconditional ImageNet in (Chen et al. 2019) with equivalent batch size and compute resources. Furthermore, score-based models and diffusion models have not been shown to scale to unconditional ImageNet at the 128x128 resolution (the largest unconditional models we are aware of use size 64x64), which our EBM does at a reasonable budget.
>
> * *CIFAR10 and has worse natural accuracy ... and ImageNet results are clearly not outperforming baselines*: Please see our general response about our updated CIFAR-10 results. After further verifying our best results, we find that the EBM can achieve SOTA defense not only among purification methods, but among all methods. Furthermore, our method is the only purification method that can secure ImageNet classifiers. This is a fundamentally different approach than adversarial training and an important proof of concept for the scalability of EBM defense. While countless purification-based defenses have been broken, the EBM defense can withstand SOTA attack methods. This is a unique example of symbiosis between supervised and unsupervised modeling that is especially suitable for the EBM. For example, score models are not able to provide defense using the same technique (see Appendix G).
>
> * *There are no density estimates on the test set, only FID*: Please refer to our general response on log likelihood. One can model a density without estimating the partition function. In particular, log likelihood is unable to provide information about whether the probability mass of a density model concentrates on realistic images or on unrealistic images. This subtlety has been a blind spot in evaluation of generative models for quite some time, and our work brings much-needed attention to the surprising challenges of proper density modeling. We strongly believe density models should be evaluated primarily based on where the model assigns its probability mass and that this is best investigated by performing longrun sampling. An under-recognized strength of the EBM among other generative models is the ability to learn a well-formed potential energy surface.
>
> * *the amount of compute and time needed to perform training and inference with EBMs*: We will include an appendix of approximate training and experiment times in future versions. All models were trained on either a TPUv2-8 or TPU3-8 and took no longer than 3 days to complete. For each experiment we utilized at most 8 TPU cores, which is a moderate amount for current generative modeling research. We further note that EBM learning is highly parallelizable up to the limit of one chain per core, and that training and inference speeds up as more cores as used. Finally, shortrun image synthesis with an EBM at inference is much less costly than with diffusion or score models.
>
> * *The paper does not have many technical novelties*: Please refer to our general response regarding novelty.
>
> * *a reasonable alternative is to use a good "rejuvenation model" and avoid EBM entirely*: Such an approach would very likely yield nonviable results for defense or density modeling due to the instability of sampling paths from the learned EBM. Alternatives such as variational approximation yield much worse distributional approximations than MCMC, and even MCMC must be pushed to its limit to reach the model steady-state. Training with midrun or longrun MCMC sampling (or simulating these trajectories) is the simplest and most effective tool for developing robust landscape structures, which is one of the crucial points of our work. On the other hand, rejuvenation improves sample quality. Our methods strike different balances between these objectives for shortrun (fast rejuvenation), midrun (moderate rejuvenvation), and longrun (infrequent rejuvenation) learning.

---

### Official Review · Reviewer_gMv3 · 2021-11-02

**Correctness:** 3
**Technical Novelty And Significance:** 3
**Empirical Novelty And Significance:** 3
**Recommendation:** 8
**Confidence:** 4

**Main Review:**

Image synthesis section comments:
- It should be more clearly explained why batch-normalization in the generator isn’t enough to fix the issue.
- Section 2.2 is confusing. Isn’t the persistent bank just over the distribution (x, z) factorized as p(x|z)p(z), where z is the latent vector and x is the generated image?

Adversarial defense
- “Annealing” in the final paragraph of 3.1 refers to “annealing the learning rate”, right?
- Section 3.2 it’s unclear how the hyperparameters are specified. As I understand it, two of K, K_def, and p_rejuv must be specified. Which of the two are specified, and to what values?
- Are the experiments justifying the reasoning given in the last paragraph of 3.1? How similar are the trajectories of length K late in training to actual trajectories of length K_def, when the learning rate is annealed?
- In Figure 6, why does robustness decrease for large K? Shouldn’t larger K used during training lead to better performance when using K_def at test-time?
- How do the results compare when using data samples instead of a pre-trained generator?

Density Estimation
- First paragraph, Section 4.1
   - “However, persistent learning without rejuvenation has shortcomings mentioned in Section 3” -> Section 2?
- “Persistent samples that are newly rejuvenated (up to about 50K Langevin steps since rejuvenation, and possibly many more) cannot be approximate steady-state samples for any current known rejuvenation sources, including data, generators, and noise.”
- I think this sentence needs more elaboration / definition of terms. My understanding is as follows: Samples in a bank that have undergone less than 50k langevin updates cannot be steady-state samples
- What is 50k defined relative to? Is it supposed to be half of 100k, which is the number of steps being used to estimate the density?
- What is the definition of “lifetime” Langevin updates? Does this refer to the number of Langevin updates applied to a sample between different training iterations (e.g. by being sampled from the persistent bank and its updated version back into the persistent bank)
- “Samples in the newly rejuvenated bank that have been updated sufficiently many times will eventually replace samples from the bank used to update the EBM, at which point newly rejuvenated states will be added to the first bank.”
  - “the first bank” refers to the bank for newly rejuvenated samples, correct?
- How do the results compare when using data samples instead of a pre-trained generator?
- Are there experiments measuring the quality of the EBM density?
  - My understanding is that these experiments show that EBMs can be trained so that long-run MCMC samples (which are closer to samples from the true EBM density) are high-quality. Are there experiments measuring how good the density is directly? For example, experiments on out-of-distribution detection? A small tractable model on MNIST where the normalizing constant can be estimated, and compare the log-likelihood to exact density models, as well as exact samples to long-run MCMC samples?
- The prior EBM used in eqn. 2 seems like a little bit of a hack. To me it sounds like it could just be “slowing-down” how fast MCMC converges before oversaturation. Are there any plots showing that MCMC achieves a wide-region of stability where the sample quality and diversity stays relatively constant?

Typos:
- Section 1 (Introduction)
  - “an misaligned” (3rd paragraph)
- Section 2.2 (Hybrid Persistent Cooperative Initialization)
  - “uses paired latent and image states that a drawn from” (Figure 2 caption)
- Section 4 (Longrun Sampling for Density Estimation)
  - “these outcomes not equivalent” (2nd paragraph)
  - “the lack scalable methods” (2nd paragraph)
  - “by introducing an MCMC initialization can incorporate” (2nd paragraph)
- Section 4.1 (Incorporating Rejuvenation in Density Estimation)
  - “initializatin” (Figure 4 caption)
  - “remain the the” (Figure 4 caption)
  - “burnin bank” (Figure 4 caption)
  - “until they have approach” (Figure 4 caption)
  - text under eq. (2) does not explain U_0 term

**Summary Of The Paper:**

The paper describes techniques to train EBMs according to the length of MCMC trajectories required. The paper explores three applications of EBMs: image synthesis, adversarial defense, and density estimation. Each of these applications uses a different regime of MCMC sampling length. An approach is described for modifying the initialization of MCMC both during training and at test-time to improve the performance in each of these applications.

For image synthesis, the paper proposes using a combination of persistent chains and cooperative learning. Persistent chains with rejuvenation can cause instability since negative samples can vary between freshly rejuvenated samples and longer run samples. However, no rejuvenation often leads to lack of sample diversity. Persistent banks don’t scale well to large datasets since they cannot capture all the variability of the larger dataset. Instead, chains can be rejuvenated from a fast generator model, which is trained jointly with the EBM, known as cooperative learning. The authors identify and fix a key issue with cooperative learning. The issue is that generators early in training are unable to generate a diverse set of samples that the EBM can refine. One fix is to use batch normalization in the generator. Instead, the authors propose using a persistent bank over the joint distribution of latent noise and generated images (where the latent noise is input to the generator), which the sampling process is rejuvenated from with the usual process.

**Summary Of The Review:**

Overall the paper is well-structured and clear, and presents unique and interesting ideas for training unconditional EBMs for different applications. There are some places where clarity could be improved, and some additional experiments which I think might improve some of the points in the paper.

---

> ### Author Response · Authors · 2021-11-20
> **Response to Reviewer gMv3 [1/2]**
>
>
> We appreciate your time and your supportive evaluation of our work. We agree that the primary strength of our work is a unique perspective on the practical aspects of EBM learning that leads training strategies for different applications. Your suggestions will help us improve our work.
>
> * *Why batch-normalization in the generator isn’t enough:* Empirically, the amount of diversity that batch norm adds is very limited, as shown in Figure 5. Analysis of batch norm is beyond the scope of this paper, but it appears that a latent $z$ defines a rough skeleton for the image $g(z)$ that can undergo limited modification from batch pairings with other $z'$.
> * *Isn’t the persistent bank just over the distribution (x, z) factorized as p(x|z)p(z)*: The theory in Section 2.2 directly follows the Cooperative Learning framework (Xie et al. 2018). The inclusion of the persistent banks do not change the Maximum Likelihood derivation for either the EBM or generator. The persistent banks can be understood as samples from $p(x, z)$ factorized as $p(x;\theta)p(z|x;\phi)$ in the theoretical formulation. However, we know in this section the EBM will be non-convergent so the $x$ bank will not actually represent $p(x;\theta)$. Further, we do not update the $z$ vectors to follow $p(z|x;\phi)$. This is consistent with the original Cooperative Learning code implementation, which also does not update the $z$ vectors. The justification is that MCMC sampling on the $z$ vectors is approximately the identity mapping when the model is well-learned. We are stretching this assumption slightly further than the original Cooperative Learning framework since our $x$ samples travel further from their generator initialization, but not that much further since we rejuvenate samples very rapidly with a 50% probability. We note the line between theory and practice is unclear for all non-convergent EBMs, not only our method, and in this section we focus on practical aspects needed for good synthesis.
> * *“Annealing” refers to “annealing the learning rate”, right?*: Yes, we only anneal the learning rate of the optimizer. There is no temperature or step size annealing for MCMC sampling trajectories. All MCMC sampling is performed at a fixed temperature that is equal to or very close to the training temperature.
> * *Section 3.2 it’s unclear how the hyperparameters are specified*: $K$ refers to the number of shortrun MCMC steps during training, which we set to at most $100$ across the paper. $p_\text{rejuv}$ is the rejuvenation rate during training, which is 0.05. $K_\text{def}$ refers to the number of defense steps used during defense evaluation after training has concluded. The midrun training is designed so that $K_\text{def}$ can be any value less than or equal to $K / p_\text{rejuv}$. $K_\text{def}$ must be at least several hundred steps for successful defense.

---

> > ### Author Response · Authors · 2021-11-20
> > **Response to Reviewer gMv3 [2/2]**
> >
> >
> > * *Are there experiments justifying the reasoning given in the last paragraph of 3.1?*: Yes, we included this experiment in the supplementary material and will include in a future appendix. Annealing is needed to stabilize the midrun trajectories.
> > * *why does robustness decrease for large K?*: This happens because natural accuracy decreases for large $K$, since sampling for longer disrupts the original image class features. Decrease in natural accuracy leads to decreased robust accuracy even if sampling for longer removes more adversarial signals.
> > * *How do the results compare when using data samples instead of a pre-trained generator?*: We will include these results in future versions for both midrun and longrun learning. The differences are negligible.
> > * *Persistent samples that are newly rejuvenated (up to about 50K Langevin steps since rejuvenation, and possibly many more) cannot be approximate steady-state samples*: Your understanding is correct. Intuitively, current rejuvenation sources always generate samples very "far" from the steady-state, even if they look quite similar to steady-state samples. Steady-state convergence can only be achieved after a long burn-in process (empirically, at least 50K steps for our method. Other methods/MCMC samplers might require more).
> > * *What is 50k defined relative to?*: 50K Langevin steps represents an empirical upper bound on the length of the burn-in trajectory from a generator rejuvenation needed before a sample is an approximate steady-state sample (or at least a locally stable sample). In the longrun section, models are only updated with samples that have undergone 50K MCMC updates since rejuvenation. This prevents the misalignment effects that can result from including newly rejuvenated MCMC samples in the EBM weight update.
> > * *What is the definition of “lifetime” Langevin updates?*: Your understanding is correct. The "lifetime" of a sample during training refers to the total number of MCMC steps that a sample undergoes between the time it is newly introduced via rejuvenation and the time that it is overwritten by a newly rejuvenated state.
> > * *"the first bank” refers to the bank for newly rejuvenated samples?* Yes.
> > * *Are there experiments measuring the quality of the EBM density?* Please refer to the general response to all reviewers. For the reasons discussed, to our knowledge longrun MCMC sampling is the most rigorous method to test the placement of the model's probability mass. Log likelihood is surprisingly unable to detect the widespread but often unrecognized phenomenon of steady-state misalignment. Given the difficulty of learning a valid steady-state, we focus our efforts in this section on the learning process itself.
> > * *The prior EBM used in eqn. 2 seems like a little bit of a hack*: This approach is based on the intuition that it should be easier to learn a convergent EBM by patching up the "leaks" of a non-convergent EBM, rather than learning a convergent EBM from scratch. In other words, we use the second EBM to learn a residual energy function needed to correct the first energy. While one might doubt its elegance, the method is theoretically valid and has precedence in techniques like residual layers.

---

> > > ### Comment · Reviewer_gMv3 · 2021-11-21
> > > **Response**
> > >
> > > Thanks for the very thorough response to my review, and the additional comments and supplementary information added for all reviewers. I think this address all of my concerns.

---

### Author Response · Authors · 2021-11-20
**Revision Changes**

For this initial response, we have updated our submission with a Supplementary Appendix (in the Supplementary Materials) that helps address some reviewer questions. We will incorporate these revisions into our paper before the response period ends. Here we address noteworthy changes in the content of our paper that will be included in future versions.

**Changes to Paper Results**

* In Table 2 of the original submission, aside from the typo, the CIFAR-10 defense result is not representative our of best results. Shortly before submission, we had reason to suspect that there was an implementation error in the experiments for our best CIFAR-10 defense. We decided to report a more conservative result that had been previously verified as rigorously as possible. Since submission, we have re-verified our best result and will include this score in future versions. In particular, we achieve 67% adversarial accuracy and 86% natural accuracy on the CIFAR-10 test set for the same settings in the paper. Our improved results do not involve any training changes in the EBM, the only difference is use of a better naturally trained classifier and more careful selection of an EBM checkpoint from a collection of saved checkpoints during training.

* In the longrun section, the paper states that the prior EBM providing an initial landscape for longrun learning should be a midrun EBM. This is mistaken. We find that it is better to use a shortrun EBM as the prior because it accelerates exploration into oversaturated regions so that the new EBM can seal the leaky energy basins and stabilize longrun sampling. The ImageNet and Celeb-A experiments in the original submission use a shortrun EBM as the prior EBM and the CIFAR-10 experiments use a midrun EBM as the prior EBM. We achieved better stability by using a shortrun prior EBM for CIFAR-10 and will include this result in future versions. We will include a more thorough discussion of this choice in future versions.

---

> ### Author Response · Authors · 2021-11-23
> **Paper Revisions for Response Period**
>
> We have updated our paper to include revisions from the reviews. We again thank all reviewers for their engaged and thorough assessment of our work. The revisions prompted by the discussion have helped us to significantly improve our work. A brief summary of the changes are as follows:
>
> * Figure 1 has been changed to reflect the new CIFAR-10 longrun EBM. FID scores approximately stabilize after as many as 1M steps. This trajectory magnitude pushes the limits of what it is possible to feasibly evaluate.
> * Included the CIFAR-10 defense scores in Table 2 as discussed in the author responses.
> * Included 1M step FID scores for CIFAR-10, Celeb-A, and ImageNet.
> * Replaced the shortrun FID scores with the FID scores from models that do not use Langevin gradient clipping, as discussed in the author responses.
> * Included a brief appendix summarizing the role of the prior EBM for longrun learning
> * Added ablative study showing the importance of learning rate annealing
> * Added tables for each training and evaluation experiment to the paper.
> * Included discussion about log likelihood vs. steady-state sampling and probability mass distribution
> * Supplemented longrun analysis of score models with longrun analysis of flow and diffusion models
>
> We believe this covers most of the points touched upon in reviews that are within our ability to address within the response period. We are still working on performing experiments to test the effect of data initialization compared to generator initialization for midrun and longrun learning. This study is important and will be included in future versions. However, we are not necessarily claiming that generator distribution is better that data initialization, merely that it is an efficient in-distribution initialization that leads to a self-contained generative model for initializing an EBM. Exhaustive experiments are quite expensive and we decided to remain within the generator framework due to computational limitations and the preliminary studies showing no difference between the generator and data initialization methods.

---

### Author Response · Authors · 2021-11-20
**General Response to Reviewers**


We would like to thank all reviewers for their time and for their valuable insights. Incorporating points made in the reviews will help us improve the presentation of our work and the help us clarify problems we aim to address. This general response addresses questions raised by multiple reviewers about the density modeling evaluation and overall novelty.

**Evaluating Density Models**

Several reviewers raised questions about our density modeling experiments. In this work, we are primarily interested in learning density models that assign the majority of probability mass in realistic regions of the image space. This goal is, surprisingly, distinct from the goal of likelihood maximization. In particular, one can achieve high likelihood with mixture models where only an infinitesimal portion of mass is assigned to a mixture component that approximates the true density, while the majority of probability mass is assigned to a degenerate distribution (see [a], "Great Log Likelihood but Poor Samples"). The steady-state of MCMC sampling with such a mixture distribution would concentrate on the degenerate distribution even though the structures of the true density exist in high-energy regions. Therefore, log likelihood cannot detect if a model has assigned probability mass in realistic region of the image space. Recent observations have shown that this situation is not just hypothetical (Nijkamp et al. 2020). Non-convergent EBMs are practical examples, since these models consist of a mixture of partially formed high-energy energy basins enabling effective shortrun synthesis in realistic regions of the image space and much lower-energy basins in unrealistic image regions that dominate the probability mass. The same misaligned landscape structure extends beyond EBM to RBMs [b], score models (see Appendix G), normalizing flows, and diffusion models (see Supplementary appendix, to be added to future versions). Log likelihood can only indicate the presence of energy landscape features that are similar to the ground truth landscape, but it cannot detect whether these features will "leak" into lower-energy basins that represent the true mass distribution. Like shortrun sample quality, high log likelihood is often misleading false evidence that a model density concentrates on realistic images.

Overall, we believe that FID using true samples from the EBM and data samples is an appropriate, if rough, measure of successful density modeling. The subtlety involves generating true samples from the EBM. We believe that longrun MCMC sampling is the most practical and principled tool for obtaining approximate samples from the EBM steady-state, and our investigations show that density modeling with properly calibrated probability mass is currently most effectively accomplished using MCMC-based EBM training. Though we use the FID metric, our density modeling experiments are best described as "realism preservation" experiments rather than "image generation" experiments.

[a] A note on the evaluation of generative models. Thiel et al. ICLR 2016. https://arxiv.org/pdf/1511.01844.pdf

[b] Equilibrium and non-equilibrium regimes in the learning of Restricted Boltzmann Machines. Decelle et al. NeurIPS 2021. https://arxiv.org/pdf/2105.13889.pdf

**Novelty of Technical Contributions**

Our work has several key technical innovations. Merging cooperative and persistent initialization to improve initialization diversity is non-trivial and we tried different variations of this concept, most of which yielded poor results. The shortrun method presented in this paper represents a simple and effective synthesis, and the results scale surprisingly well. By introducing pretrained-generator based rejuvenation during training we dramatically improved EBM adversarial defense results in our mid-run model. Our longrun sampler and experimental evaluation represent a significant departure from mainstream approaches for learning and evaluating density models. We believe the community will benefit from a more thorough analysis of the widespread and under-recognized problem of steady-state misalignment. Learning an aligned steady-state while still including rejuvenation in the learning process is an important step towards scalable and principled density modeling with EBMs.

---

### Decision · Program_Chairs · 2022-01-20

**Decision:**

Reject

**Comment:**

This paper proposed a strategy to train EBMs according to the length of MCMC trajectories required. The paper covers three settings with the different length of MCMC: image synthesis, adversarial defense, and density estimation. The reviewers generally find that there are interesting ideas and promising results in the paper, but the paper is not ready to publish at its current stage. The argument regarding density estimation and FID evaluation is not convincing. The proposed method is also more complicated than the baseline methods (CoopNets and PCD), and we would need a stronger argument for the added complexity.